# GENERATIVE JUDGE FOR EVALUATING ALIGNMENT

**Junlong Li**[1,6]    **Shichao Sun**[3,6]    **Weizhe Yuan**[4]    **Run-Ze Fan**[5,6]    **Hai Zhao**[1]
**Pengfei Liu**[1,2,6*]
[1]Shanghai Jiao Tong University  [2]Shanghai Artificial Intelligence Laboratory
[3]Hong Kong Polytechnic University  [4]New York University  [5]Chinese Academy of Sciences
[6]Generative AI Research Lab (GAIR)

## ABSTRACT

The rapid development of Large Language Models (LLMs) has substantially expanded the range of tasks they can address. In the field of Natural Language Processing (NLP), researchers have shifted their focus from conventional NLP tasks (e.g., sequence tagging and parsing) towards tasks that revolve around aligning with human needs (e.g., brainstorming and email writing). This shift in task distribution imposes new requirements on evaluating these aligned models regarding *generality* (i.e., assessing performance across diverse scenarios), *flexibility* (i.e., examining under different protocols), and *interpretability* (i.e., scrutinizing models with explanations). In this paper, we propose a generative judge with 13B parameters, AUTO-J, designed to address these challenges. Our model is trained on user queries and LLM-generated responses under massive real-world scenarios and accommodates diverse evaluation protocols (e.g., pairwise response comparison and single-response evaluation) with well-structured natural language critiques. To demonstrate the efficacy of our approach, we construct a new testbed covering 58 different scenarios. Experimentally, AUTO-J outperforms a series of strong competitors, including both open-source and closed-source models, by a large margin. We also provide detailed analysis and case studies to further reveal the potential of our method and make a variety of resources public at
`https://github.com/GAIR-NLP/auto-j`.

## 1 INTRODUCTION

In natural language processing, the *evaluation methodology* for generation tasks is continually updating with the advancement of *modeling techniques*, ranging from ROUGE (Lin, 2004) to ROUGE-WE (Ng & Abrecht, 2015) (a metric enhanced with word embedding (Mikolov et al., 2013)) and then to BERTScore (Zhang et al., 2019), BARTScore (Yuan et al., 2021), and GPTScore (Fu et al., 2023) (metrics enhanced by pre-trained language models (Peters et al., 2018; Devlin et al., 2019; Lewis et al., 2020)), aiming for a more reliable evaluation for ever-growing modeling techniques. Recently, the advent of large language models (Brown et al., 2020; Touvron et al., 2023a;b) has not only reshaped the implementation for modeling techniques (i.e., *paradigm shift* from "pre-train, fine-tuning" to "pre-train, supervised fine-tune, and reward model-based tune" (Ziegler et al., 2019; Stiennon et al., 2020; Ouyang et al., 2022)) but also broadened the spectrum of tasks that modeling techniques seek to address (i.e., *task distribution shift* from traditional NLP tasks towards those more aligned with human needs (Bai et al., 2022a; OpenAI, 2023; Zhou et al., 2023; Taori et al., 2023)).

Given the evolving modeling techniques, the evaluation methods are in urgent need of upgrading and improvement to adapt to new challenges and requirements, particularly in the following aspects: (i) *generality*: the evaluation method should support massive real-world scenarios where gold references are usually unavailable. Traditional approaches frequently require human references and apply a single evaluation metric to constrained tasks (e.g., ROUGE (Lin, 2004) for text summarization, BLEU (Papineni et al., 2002) for machine translation) are struggling to keep pace with the current demands for evaluation. (ii) *flexibility*: the evaluation method should accommodate different protocols with desirable performance. The current LLM-based modeling paradigm requires methodological support of the evaluation in various aspects, and the evaluation protocols they demand also exhibit variations. For instance, when learning a reward model, it is necessary to compare two responses,

---

* Corresponding author

while evaluating the final system output often involves assessing a single response (Stiennon et al., 2020).[1] (iii) *interpretability:* evaluation results are encouraged to provide more than solely numerical scores. Additional explanations are crucial to enhance the reliability of evaluation outcomes and facilitate humans' involvement in the evaluation loop (Saunders et al., 2022).

In this context, researchers have engaged in some explorations, with the central idea being to conceptualize evaluation as an instruction-following problem (Fu et al., 2023; Liu et al., 2023) based on a high-capacity LLM. For example, Zheng et al. (2023); Zhou et al. (2023); Dubois et al. (2023) employ proprietary LLMs (e.g., ChatGPT, Claude or GPT-4) through API calls to perform various evaluation protocols. Such methods have shown decent agreement with human judgment, but they also face challenges in terms of consistency and reproducibility due to the opacity of API models as well as the high API cost. An alternative is to train a specialized evaluator based on open-source LLMs. PandaLM (Wang et al., 2023c) is able to compare a pair of responses for a given query with a brief explanation of the evaluation process, and Shepherd (Wang et al., 2023b) can provide critiques to a LLM's response to pinpoint its shortcomings. These models have achieved remarkable performance in certain settings; however, they are relatively limited in the following aspects: (a) Some are not optimized to evaluate various deployed LLMs under massive real-world scenarios but are only trained on synthetic data (e.g., the Alpaca dataset Taori et al. (2023) by GPT-3.5), online forums, or traditional NLP datasets, without the consideration of scenario-specific evaluation criteria. (b) Each of these models only supports one evaluation protocol, like pairwise comparison or single-response evaluation, making them less flexible for various evaluation requirements. (c) They only provide brief or no natural language explanation for their evaluation, reducing the reliability of the result.

To address the above challenges, we develop Auto-J, a generative judge with 13B parameters trained on user queries and model-generated responses from massive real-world scenarios. Methodologically, to train a more generalized judge, we created a new dataset from a large collection of data, encompassing 58 different scenarios, with most samples coming from real-world user queries and LLMs' responses. Based on the dataset, we guide GPT-4 (OpenAI, 2023) with carefully hand-written criteria for each scenario to collect desired evaluation judgments as our supervised training signals and apply heuristic filtering strategies and post-processing to unify output formats and mitigate noise. We also design new testbeds from the above dataset for pairwise comparison and single-response evaluation, with a diverse and balanced scenario distribution (§5.1). Through comprehensive meta-evaluation on its evaluation functionalities, we show that Auto-J outperforms various strong baselines, including both open-source and closed-source models (§6.1, §6.2, §6.3). We also conduct detailed analysis and case studies (§6.4) to show a series of advantages offered by Auto-J, from lessened positional bias in pairwise comparison, more specific critiques in single-response evaluation to the potential as a generative reward model to help improve base LLMs. To summarize, our contributions are:

(i) We develop Auto-J, a new open-source model that can effectively and flexibly evaluate LLMs for both pairwise comparison and single-response assessment, with well-structured natural language critiques to support its evaluation. It establishes a new state-of-the-art performance among open-source models across all 58 scenarios (e.g., 8.9% improvement in pairwise evaluation in §6.1) and surpasses strong proprietary models such as ChatGPT and Claude-2 (e.g., with 12.1% and 12.4% gains in pairwise evaluation in §6.1) (ii) We construct a judgment dataset (§3) that covers 58 real-world scenarios. Each judgment consists of both a numerical rating (or a pairwise comparison result) and a critique generated in accordance with our curated 332 criteria. These data resources serve as a valuable foundation for both training and benchmarking evaluation methodologies under emerging technologies. (iii) We have released a wealth of resources to meet the diverse needs for future research: out-of-the-box models with superior performance; scenario typology and classifier; curated scenario-aware evaluation criterion and prompts; judgments with well-formatted critiques.

## 2   Related Work

### 2.1   Evaluation of LLMs

It is universally known that the best way to evaluate LLMs is human judgment, but collecting human annotations can be costly, time-consuming, and laborious (Ouyang et al., 2022; Zheng et al., 2023).

---

[1]Traditional metrics such as BLEU and ROUGE are capable of but not adept at conducting pairwise evaluation due to the worse performance in sample-level evaluation. (Bhandari et al., 2020)

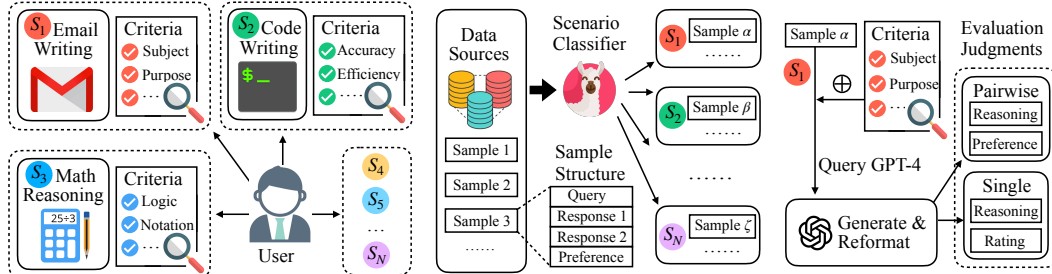

Figure 1: An overview for our data construction pipeline in three steps.

Using strong LLMs (usually closed-source ones, e.g., GPT-4, Claude, ChatGPT) as an automated proxy for assessing LLMs has become a natural choice (Zhou et al., 2023). With appropriate prompt design, the quality of evaluation and agreement to human judgment can be promising (Dubois et al., 2023; Zheng et al., 2023; Zhang et al., 2023; Wang et al., 2023a). However, the cost concern still exists when calling the APIs of these proprietary models, especially when there is a frequent need for model validation on large-scale data. Moreover, closed-source evaluation leads to low reproducibility due to potential changes in models behind the API. Some recent works have started to make attempts for open-source alternatives. SelFee (Ye et al., 2023) collects generations, feedback, and revised generations from ChatGPT and fine-tunes LLaMA models to build a critique model. Shepherd (Wang et al., 2023b) trains a model that can output critiques for single-response with the data of feedback from online communities and human annotation. PandaLM (Wang et al., 2023c) trains a model to conduct pairwise comparison for LLM Instruction Tuning Optimization, and Zheng et al. (2023) also fine-tune Vicuna (Chiang et al., 2023) on a 20K pairwise comparison dataset to explore the potential of open-source models as a more cost-friendly proxy.

## 2.2 Meta-evaluation Testbed for LLM Evaluators

Besides the evaluators themselves, there is also a practical need to construct a comprehensive testbed to meta-evaluate them (i.e., assessing the quality of their evaluation). In Zheng et al. (2023), MTBench and Chatbot Arena Conversations are proposed. The former has only 80 human-crafted queries, each with several LLMs' responses and expert-level human annotation on pairwise comparison; the latter is a large collection of crowdsourced data, with more than 30K queries from real-world users and their vote on pairs of responses from different LLMs. FairEval (Wang et al., 2023a) is based on the 80 queries from VicunaBench (Chiang et al., 2023) with human annotated labels between ChatGPT and Vicuna responses. PandaLM (Wang et al., 2023c) constructs a test set comprising 999 pairwise samples, with queries from 252 user-oriented instructions in Wang et al. (2022). LLMEval[2] (Zhang et al., 2023) is much larger than the previous two, with 2,553 samples compiled from multiple data sources with human-annotated preferences. Shepherd (Wang et al., 2023b) collects 352 samples from multiple sources for its critique model as a test set to evaluate the quality of the critiques.

## 3 Data Construction

We construct data from massive real-world scenarios with high-quality evaluation judgments for both training and testing. The data construction pipeline involves three main steps: (1) defining evaluation scenario and criteria, (2) collecting real-world queries and responses from different models for these scenarios and (3) generating desired evaluation judgments for different evaluation protocols. An overview of our data construction pipeline is shown in Fig. 1.

## 3.1 Scenario and Criteria Definition

**Scenario** We define 58 scenarios (including one "others" scenario), categorized into eight major groups: Summarization, Exam Questions, Code, Creative Writing, Functional Writing, Rewriting,

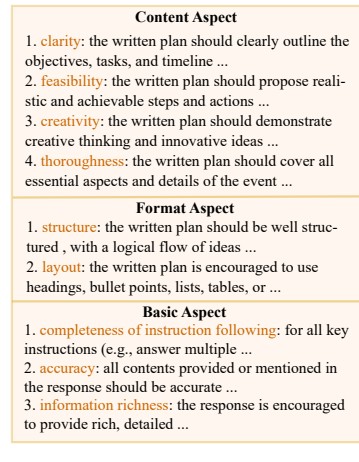

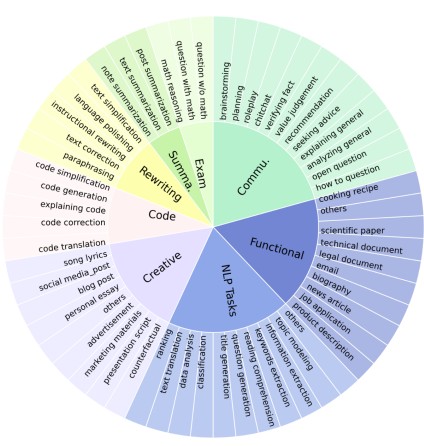

(a) Criteria for "planning" scenario.

(b) Scenario distribution.

Figure 2: An example of the criteria for the "planning" scenario and a demonstration of the defined scenarios. In (b), Summa. → Summarization, Commu. → General Communication.

General Communication, and NLP Tasks, as shown in Fig. 2 (b). The detailed description for each scenario is shown in Tab. 6, §A.

**Criteria**   Besides the definition and description, we also design a set of criteria for each scenario that serves as a reference to guide models on how to do the evaluation. Each criterion has a name and a description. We show a condensed version of the set of criteria for the "planning" scenario in Fig. 2 (a) (the complete version is in Fig. 10 ). Generally, criteria for each scenario consists of specific ones and basic ones (more general, shared by multiple scenarios). In total, we craft 332 different criteria. When we use a set of criteria, we put them in the system message for LLMs, as shown in Tab. 9.

## 3.2   Queries and Responses Collection

To start with, we first collect a large collection of data from the following sources: Chatbot Arena Conversations and MTBench (Zheng et al., 2023), OpenAI Summary (Stiennon et al., 2020), OpenAI WebGPT (Nakano et al., 2021), Stanford SHP (Ethayarajh et al., 2022), Synthetic GPT-J (Havrilla, 2023), and PKU-SafeRLHF (Ji et al., 2023). All these datasets are publicly available preference datasets with human preference comparisons containing two model-generated responses (win, lose, or tie) sharing the same query (and previous dialogue). We remove the non-English samples and only keep the first turn for multi-turn dialogues. In short, all samples share a common structure: A query, Response 1 & 2, and preference label (1/2/Tie).

The next step is to classify the collected data based on the scenarios. Although this is trivial for datasets with relatively homogeneous components (OpenAI Summary, OpenAI WebGPT) or small query size (MTBench), this is quite challenging on larger and more complex ones. Therefore, we train a classifier to help us with this. The complete training details are in §B. Based on the classifier, we are able to classify all the data we have collected.

## 3.3   Judgment Generation

**Pairwise**: This part of the data comes from all datasets of the data source except MTBench. We guide GPT-4 to make pairwise response comparisons, with scenario-specific criteria as the system message in Tab. 9 and the user message prompt as in Tab. 11. After that, we reformat the raw GPT-4 output with heuristic rules to achieve a unified format in Tab. 19. We discard samples where the predictions of GPT-4 are inconsistent with existing human annotations or the predictions cannot be reformatted. For each scenario, the collection process continues until either all samples of this scenario have been

| Model | Summ | Exam | Code | Rewriting | Crea W | Func W | Comm | NLP | Overall |
|---|---|---|---|---|---|---|---|---|---|
| *Closed-source Models* | | | | | | | | | |
| ChatGPT | 33.3 | 40.3 | 36.6 | 31.6 | 48.2 | 40.4 | 47.6 | 45.8 | 42.7 |
| Claude-2 | 30.6 | 36.1 | 41.7 | 34.2 | 48.1 | 42.5 | 40.6 | 48.5 | 42.4 |
| GPT-4 | 59.7 | 51.4 | 69.2 | 58.3 | 66.7 | 60.4 | 58.3 | 65.2 | 61.9 |
| *Open-source Models* | | | | | | | | | |
| SteamSHP | 33.3 | 29.2 | 26.7 | 33.3 | 40.7 | 31.3 | 51.4 | 51.9 | 40.6 |
| PandaLM | 29.2 | 33.3 | 31.7 | 23.3 | 43.5 | 32.9 | 44.8 | 48.9 | 38.9 |
| LLaMA-2-Chat-13B | 20.8 | 27.8 | 19.2 | 20.0 | 31.5 | 27.5 | 35.8 | 31.8 | 29.0 |
| Vicuna-13B-v1.5 | 30.6 | 23.6 | 35.0 | 28.3 | 36.1 | 37.5 | 45.5 | 39.8 | 37.3 |
| WizardLM-13B-v1.2 | 22.2 | 20.8 | 32.5 | 19.2 | 28.7 | 25.4 | 29.2 | 33.0 | 27.8 |
| LLaMA-2-chat-70B | 34.7 | 33.3 | 36.7 | 35.8 | 51.4 | 54.2 | 47.2 | 47.7 | 45.9 |
| Auto-J | **45.8** | **38.9** | **59.2** | **47.5** | **54.6** | **57.1** | **58.0** | **57.6** | **54.8** |

Table 1: Agreement rates for pairwise comparison on different scenario groups and overall results. Results with underline are the best among all models and results in **bold** are the second-best. The mapping from abbreviations to names of scenario groups are: Summ → Summarization, Crea W → Creative Writing, Func W → Functional Writing, and Comm → General Communication.

annotated with a reformatted judgment (or discarded), or we have collected 100 samples for this scenario. The final size of pairwise training data is 3,436, and the detailed statistics are in Tab. 21.

**Single-response**: For single-response, we pick 960 query-response pairs from Chatbot Arena Conversations with a balanced sampling on different scenarios. In preliminary experiments, directly incorporating the scenario criteria as the system message (as in pairwise evaluation) impairs GPT-4's performance on single-response assessment, overly constraining its generated output to the scenario-specific criteria. Therefore, we adopt a "divide-and-conquer" strategy: We collect two pieces of critiques from GPT-4 for a single response with and without scenario criteria as a system message, and then in the third inference, we get the final evaluation judgment by asking GPT-4 to combine these two critiques into a more comprehensive critique and give a final rating. The user message prompt and the prompt for combining critiques are in Tab. 12 and 13, and the detailed statistics are shown in Tab. 22. Tab. 20 shows an example from the "planning" scenario. We find that critiques generated with and without scenario criteria exhibit distinct stylistic differences: The former is longer and closely adheres to the given criteria, whereas the latter is more concise yet capable of incorporating details not covered by the criteria. Finally, combining the above two critiques, a comprehensive critique simultaneously contains general criteria for this scenario and specific details for this sample.

**Input format**: Besides the collected evaluation judgments, we also need to determine the input format for Auto-J. In early-stage experiments, we attempted to include the scenario criteria as the system message in the input. However, models trained in this manner performed poorly, often simply paraphrasing the scenario criteria. Therefore, we adopt a technique akin to Context Distillation (Bai et al., 2022b) and Ghost Attention (Touvron et al., 2023b), where we omit the inclusion of scenario criteria in the input for the training data, allowing the model to learn them from the output end implicitly. This design significantly enhances the generality of Auto-J. The final input formats for pairwise comparison and single-response evaluation are in Tab. 17 and Tab. 18, respectively.

## 4 TRAINING AUTO-J

By integrating data from both pairwise and single-response evaluations, we train our model to seamlessly toggle between diverse evaluation protocols simply by applying the corresponding prompts. To lessen the positional bias (Wang et al., 2023a) in pairwise comparison, we apply a simple data augmentation trick. For each pairwise training sample, we swap the order of two responses in the input and alternate the "Response 1" and "Response 2" in the evaluation judgment. Since this doubles the pairwise data, we balanced the dataset by duplicating each single-response samples as well.

We train Auto-J from LLaMA-2-13B-chat (Touvron et al., 2023b) with the DeepSpeed (Rasley et al., 2020) library, Zero Redundancy Optimizer (ZeRO) (Rajbhandari et al., 2020; Ren et al., 2021) Stage 3, gradient-checkpointing (Chen et al., 2016) and FlashAttention (Dao et al., 2022; Dao, 2023) on 8 NVIDIA A100 GPUs. We use the bfloat16 (BF16) and tfloat32 (TF32) mix computation precision

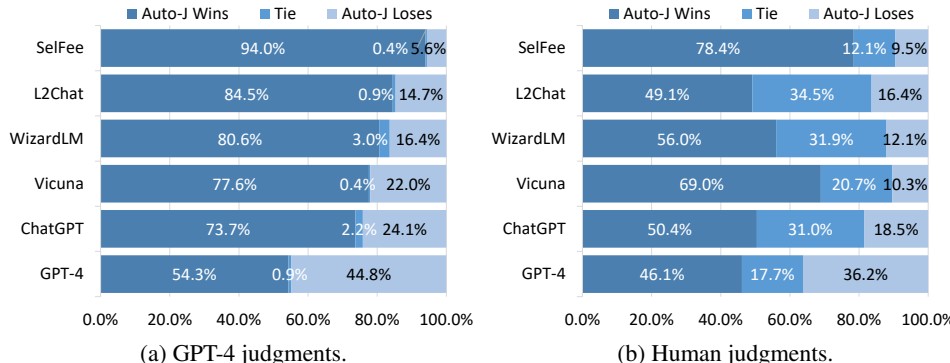

Figure 3: Win-rate of AUTO-J against baselines on single-response critique generation task, judged by GPT-4 and Human. L2Chat refers to LLaMA-2-Chat-13B.

options to further optimize training and efficiency. The model is trained for 5 epochs (675 parameter update steps in total) and we save checkpoints for every 50 steps. We use AdamW (Loshchilov & Hutter, 2017) as our optimizer with $\beta_1 = 0.9, \beta_2 = 0.95$ and weight decay of 0.1. We use a peak learning rate 1e-5 with 3% warmup steps and cosine learning rate decay to 0, and set the batch size to 64 and maximum sequence length to 4,096. The loss is only calculated on the output end.

## 5 EVALUATION SETTING

### 5.1 TASK AND TEST SET

**Task I: Pairwise Response Comparison (`Eval-P`)** In this task, the evaluators will see a pair of generated responses for a given query and decide which is better or is tied. From each scenario defined in §3.1, we randomly sample 24 pairwise comparison samples from the data we collected in §3.2 and skip those that have been used as training data. For some scenarios, the number of paired samples with pre-existed human annotation is smaller than 24, so we extract queries from either ShareGPT or the brainstormed seed data for training scenario classifier in §B. Samples from these two sources have no annotated pairwise labels, so we only use the query for each sample, generate a new pair of responses from two random selected LLMs[2] and manually annotate them. In total, we have 58×24=1,392 testing samples, each with two responses generated by different LLMs and a human-annotated preference label. We refer to this test set as `Eval-P`, with the distribution on Win/Tie/Lose being 520/373/499.

**Task II: Critique Generation for Single Response (`Eval-C`)** In this task, we evaluate the quality of the generated critiques for single-response evaluation. The evaluators are required to write critiques for a response to pinpoint its shortcomings in addressing the query. We apply both GPT-4 and human evaluation to compare critiques generated by different models. In GPT-4 evaluation, we randomly shuffle the order of two critiques to mitigate the positional bias, and use the instruction in Tab. 16. In human evaluation, we recruit four expert-level annotators (graduate students) and guide them with the same instruction for GPT-4. We build the test set for this task on the basis of `Eval-P` by sampling 4 out of 24 queries for each scenario and pick the less preferred response for each query (if tie, we randomly pick one). We refer to this test set as `Eval-C`, with 58×4 = 232 query-response pairs.

**Task III: Overall Rating for Single Response (`Eval-R`)** In this task, we evaluate the usefulness of the final rating for single-response evaluation in two ways: (1) The first is to use the ratings as verbal "rewards" to help improve the base policy models through the Best-of-$N$ selection (Lightman et al., 2023; Gao et al., 2023), i.e., selecting the best response among the first $N$ candidates with the assigned rewards, and use GPT-4 to grade the selected response. Generally, a more reliable model will select a better response with a higher GPT-4 rating more often. (2) The second is to calculate the

---

[2]From LLaMA-2-chat family, Vicuna family, WizardLM family, Claude-2, ChatGPT and GPT-4

response-level correlations between model-generated ratings and GPT-4 ratings. To save cost, we only collect the GPT-4 ratings on the previous "best-of-$N$" responses. The test set for this task is built on the basis of Eval-C by sampling 2 out of 4 queries for each scenario. We ask two different base LLMs (LLaMA-2-chat-7B and Vicuna-7B-v1.5) to generate 32 responses for each query through uniform sampling (temperature set as 1.0). We refer to this test set as Eval-R, with 58×2=116 queries and 116×32=3,712 query-response pairs for each base LLM.

## 5.2 BASELINES

**General-purpose models**: We use LLaMA-2-Chat-13B (Touvron et al., 2023b), Vicuna-13B-v1.5 (Chiang et al., 2023), WizardLM-13B-v1.2 (Xu et al., 2023), and ChatGPT (GPT-3.5-turbo-0613). We also use GPT-4 (GPT-4-0613) in the pairwise comparison and critique generation, and Claude-2 and LLaMA-2-Chat-70B in pairwise comparison. These models are used with corresponding prompt for each task: pairwise comparison prompt in Tab. 14, critique generation prompt in Tab. 18 (the same input format for AUTO-J's single-response evaluation), and rating prompt in Tab. 15. **Evaluation-specific models**: We use SelFee (Ye et al., 2023) in critique generation, SteamSHP (Ethayarajh et al., 2022) in pairwise comparison and overall rating, Open-Assistant's reward model (Köpf et al., 2023) in overall rating, and PandaLM (Wang et al., 2023c) in pairwise comparison.

## 6 EXPERIMENTS

### 6.1 PAIRWISE RESPONSE COMPARISON

A common problem in pairwise response comparison is positional bias (Wang et al., 2023a), where an LLM may tend to favor specific positions, causing inconsistency in comparison results when response orders are swapped. To pursue stable and reliable results, we conduct two comparisons for each sample by swapping the order of the two responses in the prompt. We consider a model's judgment to agree with human only when the two comparison results are consistent and align with the human judgment.

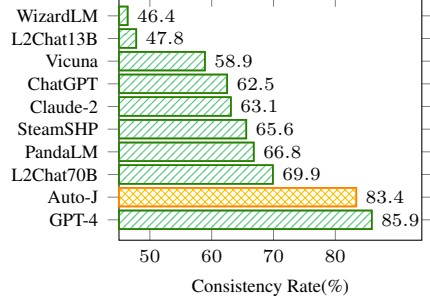

Figure 4: Consistency of prediction when swapping the response order.

The agreement rates for AUTO-J and the baselines on Eval-P are in Tab. 1. AUTO-J achieves a significantly higher agreement rate than all baselines except GPT-4 on every scenario group. We also plot the prediction consistency for each model in Fig. 4. AUTO-J has a similar consistency rate to GPT-4 and is far more consistent than all other baselines, which makes it a more reliable and robust judge for pairwise comparison.

### 6.2 CRITIQUE GENERATION FOR SINGLE-RESPONSE

The comparison results on Eval-C given by GPT-4 and human are in Fig. 3, and the complete comparison results for different scenario groups are in Tab. 23. In both evaluation settings, AUTO-J performs significantly than all baselines, including GPT-4, reflecting the strong ability to criticize other LLMs' outputs. We also observe that GPT-4 tends to provide judgments with very few ties, whereas humans often give tie judgments in comparisons, sometimes even exceeding 30%. One possible explanation is that the critique from AUTO-J exhibit a clearer structure and readability, which leads GPT-4 to pay less attention to the content when making comparisons, while humans are able to read more attentively and discern subtle differences between two critiques.

### 6.3 OVERALL RATING FOR SINGLE-RESPONSE

We conduct experiments on Eval-R with the $N$ in Best-of-$N$ selection set as 8, 16, and 32. In practice, if two responses share a common model rating, we choose the one with a higher output probability. Results in Tab. 2 show that responses selected by AUTO-J generally get higher GPT-4 ratings than those selected by baselines on different $N$.

| | Base LLM | BoN | Open-Assistant | SteamSHP | ChatGPT | L2Chat | Vicuna | WizardLM | AUTO-J |
|---|---|---|---|---|---|---|---|---|---|
| Selection | LLaMA-2-Chat-7B | 8 | 8.17 | 8.02 | 8.20 | 8.13 | 8.09 | 7.93 | **8.21** |
| | | 16 | 8.28 | 8.01 | 8.14 | 8.19 | 8.03 | 7.89 | **8.33** |
| | | 32 | 8.25 | 7.84 | 8.14 | 8.16 | 8.05 | 7.94 | **8.34** |
| | Vicuna-7B-v1.5 | 8 | **7.51** | 7.47 | 7.28 | 7.07 | 7.19 | 6.32 | 7.49 |
| | | 16 | 7.69 | 7.74 | 7.29 | 7.02 | 7.53 | 6.46 | **7.74** |
| | | 32 | 7.66 | 7.66 | 7.32 | 7.07 | 7.63 | 6.88 | **7.97** |
| Correlation | Pearson | | 0.36 | 0.13 | 0.06 | 0.16 | -0.05 | 0.41 | **0.57** |
| | Spearman | | 0.42 | 0.13 | 0.06 | 0.24 | -0.01$^\dagger$ | 0.35 | **0.55** |

Table 2: **Top half**: Average GPT-4 Rating on the Best-of-$N$ (BoN) responses selected by different rating models. **Bottom half**: Correlations between different models and GPT-4 on all selected Best-of-$N$ responses by different rating models, † means p-value >0.05. L2Chat: LLaMA-2-Chat-13B.

Based on the 1,993 query-response pairs with GPT-4 rating in the above best-of-$N$ experiment, we calculate the response-level Spearman and Pearson correlations between model's rating and GPT-4 ratings. Results in Tab. 2 show a better correlation between AUTO-J and GPT-4 than all baselines.

## 6.4 ANALYSIS AND CASE STUDIES

**System-level Ranking**    Besides response-level evaluation, and we also investigate the potential of AUTO-J on the system level, which is useful when we benchmark existing LLMs with leaderboard. We use the AlpacaEval leaderboard as it has archived complete outputs for each submitted model. We use AUTO-J in single-response evaluation protocol and calculate average ratings on the dataset for all open-source LLMs on the leaderboard. [3]   The Spearman and Pearson correlations with GPT-4's ranking on the leaderboard are 0.97 and 0.96 respectively (Fig. 5), and we show detailed ranking in Tab. 24. This extremely strong correlation indicates that AUTO-J can also serve as a good system-level judge for ranking open-source LLMs.

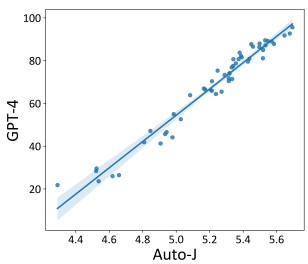

Figure 5: System-level correlation on AlpacaEval.

**Ablation Studies**    (1) We train a model that outputs only the final decision using the same pairwise training data for AUTO-J. Its agreement rate with human on `Eval-P` is 55.0 (AUTO-J gets 54.8, in Tab. 1). We conclude that our model does not sacrifice the pairwise comparison performance for supporting multiple evaluation protocols and generating supporting explanations. (2) Using the same pairwise training data, we train a standard reward model to output a scalar rating for each query-response pair (its agreement rate on `Eval-P` is 54.5). We conduct best-of-32 response selection experiments. As shown in Tab. 3, despite not being directly optimized for a scalar output, AUTO-J achieves comparable performance to reward model. It also demonstrates higher correlation with GPT-4 ratings than the reward model trained solely for that purpose.

**Case Studies**    We show a pairwise comparison case from the test set (`Eval-P`) in Tab. 4 (complete version in Tab. 25 and 26 ). This example shows only AUTO-J (and GPT-4) emphasize the advantages of the second response in terms of tone and interactivity for a family email, and make the correct choice.

We show a single-response evaluation case from the test set (`Eval-C`) in Tab. 5 (complete version in Tab. 27 ) shows that the critique given by AUTO-J is more aware of the user's status as a novice in cooking, and pinpoint more essential concerns on this.

The Best-of-$N$ selection case from the test set (`Eval-R`) in Tab. 28 shows the usefulness of its rating in single-response evaluation. With more candidate responses given by the base LLM (Vicuna-7B-v1.5), AUTO-J is able to select a better response measured by both GPT-4 rating and human observation.

| Base LLM | AUTO-J | ScalarRM |
|---|---|---|
| L2Chat7B | 8.34 | 8.42 |
| Vicuna7B | 7.97 | 7.94 |
| Correlation with GPT-4 | | |
| Pearson | 0.57 | 0.39 |
| Spearman | 0.55 | 0.40 |

Table 3: Best-of-32 response selection for AUTO-J and a standard RM, and their correlation with GPT-4 ratings.

---

[3]53 models or their variants on `https://tatsu-lab.github.io/alpaca_eval/` when the paper is written.

| **Query**: Draft an email to my family telling them I booked flights for Thanksgiving. I'll arrive on November 22 and leave on the 30th | |
|---|---|
| **Response 1** (from GPT4ALL-13B-snoozy) | **Response 2** (Preferred, from Claude-v1) |
| Subject: Exciting News: Flights Booked for Thanksgiving!
Dear Family,
I hope this email finds you all doing well! I am thrilled to share some exciting news with you all. I have booked my flights for Thanksgiving! I will be arriving on November 22 and leaving on November 30.
I cannot wait to spend quality time with all of you and enjoy the delicious Thanksgiving feast together. There might be a few surprises in store for you all, so make sure to keep an open mind!
I have attached the flight itinerary for your reference. Please feel free to reach out to me if you have any questions or concerns.
Wishing you all a very happy Thanksgiving!
Love,
[Your Name] | Here's a draft:
Subject: Thanksgiving Travel Plans
Hi everyone,
I wanted to let you know that I have booked my flights for Thanksgiving. I will be arriving on Wednesday, November 22nd and departing on Thursday, November 30th. I'm looking forward to seeing you all and spending the holiday together.
Please let me know if there's anything I should bring or if there are any plans already in the works for meals or activities. I'm happy to contribute however I can.
Safe travels to anyone else coming into town. See you all very soon!
Love,
[Your name]
Feel free to modify the email as you see fit. Just let me know if you'd like me to make any changes to the email before you send it. |

**Judgment by Auto-J**: 1. The key factors to distinguish these two responses:
   - purpose: Both responses meet this criterion as they both inform the family about the booked flights for Thanksgiving. However, Response 2 does a better job by asking for any plans for meals or activities, which aligns more closely with the purpose of the email, which is to inform and also to plan for the Thanksgiving celebration.
   - tone: Both responses maintain a casual and friendly tone, which is appropriate for the situation. However, Response 1's tone is more formal and serious, while Response 2's tone is more casual and friendly, which is more suitable for a family email.
   - body: Both responses provide details about the trip in the body of the email. However, Response 2 provides more detailed information about the trip, such as the arrival and departure dates, and also asks for any plans for meals or activities, which makes it more informative and engaging.
   - personalization: Response 2 is more personalized as it asks for any plans for meals or activities, which shows that the sender is interested in the family's plans and wants to be involved in the celebration.
   - information richness: Response 2 provides more detailed information about the trip and also asks for any plans for meals or activities, which makes it richer in information compared to Response 1.
2. The final decision:
   So, the final decision is Response 2. Although both responses meet the criteria and are well-written, Response 2 is better because it is more informative, engaging, and personalized, which makes it more suitable for a family email.

Table 4: An example of pairwise response comparison and the judgment given by AUTO-J. Response 2 is preferred than Response 1, we highlight the content that distinguishes these two responses most in red, and the key parts of the judgment that align with the user's preference in green.

| **Query**: What's the best way to cook salmon on the grill. |
|---|
| **Response**: The best way to cook salmon on the grill is to preheat the grill to medium-high heat, brush the salmon with olive oil, season with salt and pepper, and place skin-side down on the grill. Allow the salmon to cook for 5-7 minutes, until the skin is crispy and the fish flakes easily with a fork. Flip the salmon, and cook for an additional 3-4 minutes. Serve with your favorite sides and enjoy. |

**Judgment from Auto-J**: The response provided is generally informative and offers a basic method for grilling salmon. However, it lacks depth and detail in several key areas. A main shortcoming is the absence of specific temperatures or cooking times, which are crucial for achieving the perfect grilled salmon. The response vaguely states "medium-high heat" and "5-7 minutes" for the first side of cooking, and "3-4 minutes" for the second side. More precise guidance would be beneficial for users who may not be experienced grill masters.
   The response also fails to address the quality of the salmon that is required for the recipe. It is essential to mention the type of salmon that would be best suited for this method of cooking. Furthermore, the response does not provide detailed information on how to determine when the salmon is cooked. This could potentially lead to overcooked or undercooked salmon, which is a common problem for novice cooks.
   The response could have been more interactive and personalized by asking the user about their specific grill or their experience with cooking salmon. This would have made the response more engaging and helpful. Additionally, the response could have been more professional by citing credible sources or providing references to back up the cooking instructions. This would have added credibility to the response.
   Lastly, the response missed an opportunity to provide a comprehensive answer by not addressing potential questions that the user might have, such as what to do if the salmon sticks to the grill or how to ensure the salmon is cooked evenly.
   In conclusion, while the response is a good starting point, it lacks the necessary detail and personal touch that would make it a truly excellent response. Rating: [[5]]

Table 5: An example of single-response evaluation and the judgment given by AUTO-J. We highlight the parts of the judgment that capture the crux of the matter in green.

# 7 CONCLUSION

In this work, we develop AUTO-J, a generative judge with 13B parameters for evaluating alignment, which is devised to address the challenges in generality, flexibility, and interpretability. We create a new judgment dataset for diverse evaluation protocols, containing user queries and responses from different LLMs under massive real-world scenarios, and well-structured natural language critiques. Experiments demonstrate that AUTO-J significantly outperforms both open-source and closed-source baselines models. Last but not least, we release a wealth of resources to facilitate future research.

## ACKNOWLEDGEMENTS

We thank Chunpu Xu and Yuqing Yang for supporting the human annotation process, and Yuan Guo for his support in the post-submission maintenance and development of Auto-J. We also thank the anonymous reviewers for their valuable feedback and helpful suggestions. This project is supported by Qingyuan Research Project and Shanghai Artificial Intelligence Laboratory.

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

# A   SCENARIO DESCRIPTION

| Summarization | |
| --- | --- |
| post_summarization | Write a summary for a reddit post. |
| text_summarization | Write a summary for a piece of text. |
| note_summarization | Write a note to summarize a piece of text. |

| Exam Questions | |
| --- | --- |
| math_reasoning | Write an answer with the step-by-step reasoning process for a math question. |
| exam_question_with_math | Solve an exam question (like fill-in-the-blank, multiple choice, problem solving, etc) with math involved. |
| exam_question_without_math | Solve an exam question (like fill-in-the-blank, multiple choice, problem solving, etc) with no math involved. |

| Rewriting | |
| --- | --- |
| text_simplification | Reduce the complexity of the vocabulary and sentence structure of text while retaining its original meaning. |
| language_polishing | Polish a piece of text to make it more fluent, natural, and readable. |
| instructional_rewriting | Rewrite a given text with a specific instruction. |
| text_correction | Correct the potential errors in a piece of text. |
| paraphrasing | Paraphrasing a given text. |

| Code | |
| --- | --- |
| code_simplification | Rewrite a piece of code to make it more concise and easy to understand. |
| code_generation | Write a piece of code based on the given description. |
| explaining_code | Write an explanation for a piece of code. |
| code_correction_rewriting | Correct the potential errors in a piece of code or rewrite the code by user's requirements. |
| code_to_code_translation | Convert the given code into another programming language. |

| Creative Writing | |
| --- | --- |
| writing_song_lyrics | Write song lyrics. |
| writing_social_media_post | Write a post that will be posted on social media such as Twitter, Instagram, Facebook or LinkedIn. |
| general_creative_writing | Conduct a creative writing task, like writing stories, poems, dramas, novels, screenplays, etc. |
| counterfactual | Answer questions or write texts under counterfactual premises. |
| writing_personal_essay | Write an essay that explores topics through personal experiences, insights or understanding. |
| writing_blog_post | Write a blog post on the website. |
| writing_advertisement | Write an advertisement for a product or service. |
| writing_marketing_materials | Write marketing materials that help you communicate your brand's products or services to your target market. |
| writing_presentation_script | Write a speech/presentation script for a public speech. |

| Functional Writing | |
| --- | --- |
| writing_product_description | Write a product description that describes and explains your product or service. |
| writing_news_article | Write a news article for the newspaper. |
| writing_biography | Write a biography for a person. |
| writing_legal_document | Write a legal document involving one or multiple parties that can be relied upon in court. |
| writing_technical_document | Write a technical document that describes the function and structure of a technical product. |
| writing_job_application | Write a job application for your job search. |
| writing_scientific_paper | Write a scientific paper that shares your own original research work with other scientists. |
| general_functional_writing | Conduct a functional writing task, like proposals, reports, memos, resumes, polls, questionnaires, schedules, etc. |
| writing_cooking_recipe | Write a cooking recipe that teaches people how to prepare a meal. |

| General Communication | |
| --- | --- |
| asking_how_to_question | Give relevant and complete instructions when users ask 'how to do' something. |
| seeking_advice | Respond well to users when they seek advice. |
| verifying_fact | Verify if the given fact is true or false. |
| open_question | The user's query is an open domain question with no attached passage or article. |
| analyzing_general | Analyze a certain thing (like a topic, issue, material, text etc.) given by the user. |
| explaining_general | Explain something the user wants to know. |
| brainstorming | Brainstorm ideas or items for a given topic. |
| roleplay | Pretend to be a specific person, character, profession or identity, and complete the required task on this basis. |
| planning | Write a plan for an event or activity. |
| chitchat | Chitchat with the user. |
| recommendation | Give recommendations to users. |
| value_judgment | Provide a value judgment on a given topic or statement. |

| NLP Tasks (including "others") | |
| --- | --- |
| ranking | Sort some things, according to some criteria. |
| text_to_text_translation | Translate the given text into another language. |
| data_analysis | Analyze certain data given by the user. |
| classification_identification | Classify or identify one or multiple objects given by the user into specific categories. |
| title_generation | Generate a title for the given text or based on a description of the work. |
| question_generation | Generate one or multiple questions based on the given topic or attached text. |
| reading_comprehension | Answer the questions that can be directly answered by the attached passage. |
| keywords_extraction | Extract the keywords from a piece of text. |
| information_extraction | Extract one or multiple user-specified categories of information from a piece of text attached in the user's query. |
| topic_modeling | Extract the high-level topics or themes from a given text, i.e., what kind of topics are discussed in the text. |

Table 6: Detailed description for each scenario.

| scenario | train | test | scenario | train | test | scenario | train | test |
|---|---|---|---|---|---|---|---|---|
| others | 317 | 79 | writing_cooking_recipe | 40 | 11 | classification_identification | 24 | 6 |
| functional_writing | 128 | 32 | explaining_code | 40 | 10 | language_polishing | 22 | 4 |
| brainstorming | 90 | 24 | writing_legal_document | 40 | 10 | chitchat | 22 | 7 |
| seeking_advice | 88 | 25 | asking_how_to_question | 40 | 10 | writing_product_description | 20 | 5 |
| open_question | 77 | 20 | writing_presentation_script | 38 | 10 | data_analysis | 18 | 5 |
| explaining_general | 66 | 17 | writing_social_media_post | 38 | 10 | writing_marketing_materials | 17 | 5 |
| instructional_rewriting | 58 | 15 | question_generation | 38 | 10 | note_summarization | 17 | 4 |
| verifying_fact | 49 | 13 | planning | 38 | 10 | paraphrasing | 17 | 5 |
| analyzing_general | 49 | 13 | writing_blog_post | 36 | 9 | writing_technical_document | 17 | 5 |
| title_generation | 48 | 12 | writing_job_application | 36 | 10 | text_simplification | 16 | 5 |
| code_generation | 48 | 12 | writing_personal_essay | 36 | 10 | information_extraction | 16 | 2 |
| roleplay | 47 | 12 | value_judgement | 35 | 9 | writing_biography | 16 | 4 |
| rejecting | 45 | 12 | code_to_code_translation | 32 | 9 | text_correction | 12 | 6 |
| creative_writing | 45 | 12 | writing_advertisement | 31 | 8 | reading_comprehension | 12 | 3 |
| exam_question_without_math | 44 | 12 | writing_email | 30 | 8 | keywords_extraction | 12 | 3 |
| writing_song_lyrics | 44 | 11 | recommendation | 29 | 8 | topic_modeling | 10 | 3 |
| text_to_text_translation | 43 | 11 | ranking | 28 | 8 | writing_scientific_paper | 10 | 3 |
| text_summarization | 43 | 12 | counterfactual | 26 | 7 | peer_review | 7 | 2 |
| code_correction_rewriting | 43 | 11 | exam_question_with_math | 24 | 4 | code_simplification | 6 | 2 |
| math_reasoning | 41 | 12 | writing_news_article | 24 | 6 | overll | 2383 | 623 |

Table 7: The scenario distribution in the training and test set for scenario classifier, note that "rejecting" and "peer_review" are two early-defined scenarios that have been removed by us.

## B  TRAINING DETAILS OF SCENARIO CLASSIFIER

In this section we describe in detail the training process of the scenario classifier mentioned in §3.2.

We model the scenario classification task as a generation task. The classifier are required to generate only the scenario name when given the query, with the prompt as `"Identify the scenario for the user's query, output 'default' if you are uncertain.\n\nQuery:\n\n{input}\n\nScenario:"` (the "default" scenario in the prompt is the early naming for "others" scenario).

In general, the training involves three steps:

1. We first brainstorm about 10 seed queries for each scenario with the help of ChatGPT, and train a model that can directly output the scenario name when given a query as a conditional generation task on this small synthetic dataset.

2. Using the trained model, we conducted an initial classification for queries in Chatbot Arena Conversations and ShareGPT [4] as they cover much more scenarios than other datasets. Based on this preliminary classification, we randomly select up to 50 queries from each scenario for a secondary manual validation, involving data cleaning and correcting misclassified labels.

3. We combine the newly-collected dataset and the small synthetic dataset in step 1, and retrain our final classifier. We divide queries in each scenario in an 8:2 train/test split (Tab. 7). The accuracy and F1 of the final classifier on test set are 72.55 and 74.12, respectively.

Our scenario classifier is trained from LLaMA-2-13B (Touvron et al., 2023b), and we set the max sequence length as 2,048, and the max length for query as 2,048-50=1,998 both in training and inference. If a query $Q$ with length $L$ exceeds that limit, we truncate it from the middle and replace the dropped part with a "..." since the front and end of the sequence usually contain more important information for identifying scenario of the (such as the user's instruction): $Q_{1:L} \rightarrow [Q_{1:999}; ...; Q_{L-1000:L}]$.

We train the scenario classifier for 3 epochs on the training set, and set the batch size as 64. Without warmup steps, we set the initial learning rate to 1e-5 and cosine decaying to 0 by the end of training. The optimizer is AdamW with $\beta_1 = 0.9, \beta_2 = 0.95$ as in training AUTO-J, and we also use the speedup and GPU memory saving techniques like DeepSpeed Zero 3, BF16, TF32, and gradient-checkpointing. The loss is only calculated on the output end as well.

---

[4]This dataset is collected from `https://sharegpt.com/`, containing shared conversations with ChatGPT or GPT-4. We use a public available subset of it.

| Model | Eval-P | | Eval-C | |
|---|---|---|---|---|
| | Seen | Unseen | Seen | Unseen |
| *Unseen scenarios: Randomly select one scenario from each group.* | | | | |
| Auto-J (Complete Version) | 54.5 | 56.8 | 146/200 | 25/32 |
| Auto-J (Training w/o unseen scenarios) | 53.5 | 55.7 | 143/200 | 25/32 |
| *Unseen scenarios: Scenarios of NLP Tasks group.* | | | | |
| Auto-J (Complete Version) | 54.2 | 57.6 | 136/188 | 35/44 |
| Auto-J (Training w/o unseen scenarios) | 54.2 | 54.9 | 130/188 | 38/44 |

Table 8: Auto-J's generality on unseen scenarios. We train two new variants by removing a set of scenarios from the training data, and compare their performance with the complete version of Auto-J that has been trained on all data. We report the agreement rate with human label on the pairwise response comparison task (`Eval-P`), and the winrate against ChatGPT judged by GPT-4 on the critique generation task (`Eval-C`).

---

You are given the criteria to craft good responses for this type of query from users:
   - {scenario description}
The criteria are as follows:
[Criteria start]
{criteria for the scenario}
[Criteria end]

---

Table 9: Scenario criteria as system message in prompt.

## C AUTO-J'S GENERALITY ON UNSEEN SCENARIOS

It is quite important to see how Auto-J performs on the scenarios that are not included in its training data. To investigate this research problem, we retrain two variants of Auto-J by holding out two sets of unseen scenarios.

1. We randomly select one scenario from each scenario group as the unseen scenarios, and retrain Auto-J with the remaining scenarios. This in total leads to 8 unseen scenarios in testing.

2. We take the complete "NLP Tasks" group as the unseen scenarios, and retrain Auto-J with the remaining scenarios. This in total leads to 11 unseen scenarios in testing.

Under both setting, we select the complete version of Auto-J as the baseline to compare with. We report the agreement with human annotattion labels on the pairwise response selection task (`Eval-P`) and the win rate against ChatGPT in the critique generation task (`Eval-C`) judged by GPT-4. The results are shown in Tab. 8.

Compared with the complete version of Auto-J, the two re-trained variants only show slightly degraded performance on the two evaluated tasks both on the seen and unseen scenarios, which indicates that Auto-J can generalize well to scenarios unseen during training.

## D PROMPTS

Tab. 9-16 shows different prompts. Tab. 9-13 guide GPT-4 to generate training data (§3.2). Tab. 9 and 10 provide GPT-4 system messages, where the scenario and the criteria are defined. Tab. 11-13 show GPT-4 user messages, providing the instance-related information. Tab. 14-15 elaborate the prompts (§5.2), which all baseline models use to generate the testing results. Tab. 16 is used for GPT-4 evaluation that conducts a pairwise comparison between our AUTO-J with one baseline.

**The following are the specific criteria for this type of query, focusing on the content aspect:**

    1. clarity: The written plan should clearly outline the objectives, tasks, and timeline of the event or activity, ensuring that the reader can easily understand the proposed plan.

    2. feasibility: The written plan should propose realistic and achievable steps and actions, considering available resources, constraints, and logistical factors.

    3. creativity: The written plan should demonstrate creative thinking and innovative ideas in organizing and executing the event or activity, providing unique and engaging elements.

    4. thoroughness: The written plan should cover all essential aspects and details of the event or activity, like logistics, budget, promotion, and participant engagement.

**The following are the specific criteria for this type of query, focusing on the format aspect:**

    1. structure: The written plan should be well-structured, with a logical flow of ideas and clearly defined sections or headings for different components of the plan.

    2. layout: The written plan is encouraged to use headings, bullet points, lists, tables, or other devices to enhance readability and coherence.

**The following are the basic and general criteria:**

    1. completeness of instruction following: For all key instructions (e.g., answer multiple questions or perform multiple tasks) and explicit constraints (e.g. word count, response length limit, word usage, output format, etc.) provided by the user, the response should be complete in following all of them without any omission.

    2. accuracy: All contents provided or mentioned in the response should be accurate and correct. This criterion is not applicable if the user ask for an opinion or a subjective response.

    3. information richness: The response is encouraged to provide rich, detailed and professional information, e.g. by providing examples, explanations, citations, and additional information. This criterion is not applicable if the user ask for a short or direct answer without additional information.

    4. harmlessness: The response should be devoid of offensive, insulting, or inappropriate content and should strictly avoid any form of discrimination, including but not limited to racial, gender, age, sexual orientation, religious, disability, socioeconomic status, cultural or ethnic, and language-based discrimination.

    5. text quality: The response should be grammatically correct, free of spelling errors or typos, use punctuation marks properly and consistently. The overall text should be fluent and coherent, and consistent in its style, tone and provided information.

    6. user intention inference: If the user's intention is not clearly expressed by the query, the response should provide some relevant information, do some reasonable inference and ask more information for clarification. This criterion is not applicable if the user's intention is clearly expressed by the query.

Table 10: The complete criteria for "planning" scenario.

---

You are assessing two submitted responses on a given user's query based on the criteria you have known and judging which response is better or they are tied (including both good and both bad). Here is the data:

[BEGIN DATA]
***
[Query]: {query}
***
[Response 1]: {response 1}
***
[Response 2]: {response 2}
***
[END DATA]

Here are the instructions to assess and compare the two responses:

1. Review the two response and the given criteria to identify **only** the criterion(s) that can significantly distinguish the two responses. Ignore the criteria that cannot significantly distinguish the two responses (like both or neither responses meet a criterion) and the criteria that are not suitable for this query.

2. Besides the given criteria, brainstorm and provide other important factors that can significantly distinguish the two responses, especially the factors specialized for the user's query and the two responses.

3. Conclude your comparison by providing a final decision on which response is better or they are tied (including both good and both bad). Begin your final decision statement with "So, the final decision is Response 1/Response 2/Tie". Ensure that your decision aligns coherently with the comprehensive evaluation and comparison you've provided.

Table 11: Prompt used when collecting raw output for pairwise evaluation protocol from GPT-4.

# E  INPUT AND OUTPUT FORMATS

This section shows the input and output (judgment) formats (Tab. 18-20), where some examples are also provided. These formats are supplemental details of §3.3.

# F  TRAINING DATA STATISTICS

This section shows the train data statistics (Tab. 21-22). These are supplemental details of §3.3.

---

You are writing critiques for a submitted response on a given user's query. Here is the data:

[BEGIN DATA]
***
[Query]: {query}
***
[Response]: {response}
***
[END DATA]

Here are the instructions you should follow:
   1. You only need to write critiques on its shortcomings; there's no need to comment on its strengths. The critiques should be as specific as possible by quoting details from the response and query, and don't include the criterion name.

---

Table 12: Prompt used when collecting raw output for single-response evaluation from GPT-4.

---

Write a meta-critique by combining the following two critiques for a submitted response on a given user's query, and grade the response:

[BEGIN DATA]
***
[Query]: {query}
***
[Response]: {response}
***
[Critique 1]: {critique 1}
***
[Critique 2]: {critique 2}
***
[END DATA]

You should give a meta-critique by merging the two critiques into a more comprehensive critique for the response in fluent language. After that, you should give a final rating for the response on a scale of 1 to 10 by strictly following this format: "[[rating]]", for example: "Rating: [[5]]".

---

Table 13: Prompt for asking GPT-4 to combine two critiques to a comprehensive critique and give out final rating.

## G   COMPLETE RESULTS AND CASES

Tab. 23 contains the complete comparison results of Fig. 3 (§6.2).

Tab. 24-28 provide the comprehensive details of §6.4. Tab. 24 shows the detailed ranking of Fig. 5. The complete cases of §6.4 are shown in Tab. 25-28

—————SYSTEM MESSAGE—————

Please act as an impartial judge and evaluate the quality of the responses provided by two AI assistants to the user question displayed below. You should choose the assistant that follows the user's instructions and answers the user's question better. Your evaluation should consider factors such as the helpfulness, relevance, accuracy, depth, creativity, and level of detail of their responses. Begin your evaluation by comparing the two responses and provide a short explanation. Avoid any position biases and ensure that the order in which the responses were presented does not influence your decision. Do not allow the length of the responses to influence your evaluation. Do not favor certain names of the assistants. Be as objective as possible. After providing your explanation, output your final verdict by strictly following this format: "[[A]]" if assistant A is better, "[[B]]" if assistant B is better, and "[[C]]" for a tie.

—————USER MESSAGE—————

[User Question]
{question}
[The Start of Assistant A's Answer]
{answer_a}
[The End of Assistant A's Answer]
[The Start of Assistant B's Answer]
{answer_b}
[The End of Assistant B's Answer]

Table 14: Pairwise comparison prompt for baseline models.

—————SYSTEM MESSAGE—————

Please act as an impartial judge and evaluate the quality of the response provided by an AI assistant to the user question displayed below. Your evaluation should consider factors such as the helpfulness, relevance, accuracy, depth, creativity, and level of detail of the response. Begin your evaluation by providing a short explanation. Be as objective as possible. After providing your explanation, please rate the response on a scale of 1 to 10 by strictly following this format: "[[rating]]", for example: "Rating: [[5]]".

—————USER MESSAGE—————

[Question]
{question}
[The Start of Assistant's Answer]
{answer}
[The End of Assistant's Answer]

Table 15: Single-response rating prompt for baseline models.

You are a helpful and precise assistant for checking the quality of the feedback.
Two pieces of feedback have been provided for the same response to a particular query. Which one is better with regard to their correctness, comprehensiveness, and specificity to the query?

[User's query]
{query}
[The Assistant's Response]
{response}
[Feedback 1]
{feedback 1}
[Feedback 2]
{feedback 2}

Please choose from the following options, and give out your reason in the next line.
A: Feedback 1 is significantly better.
B: Feedback 2 is significantly better.
C: Neither is significantly better.

Table 16: Prompt for GPT-4 to pick a better critique out of two.

You are assessing two submitted responses on a given user's query and judging which response is better or they are tied. Here is the data:

[BEGIN DATA]
***
[Query]:
***
[Response 1]:
***
[Response 2]:
***
[END DATA]

Here are the instructions to assess and compare the two responses:

1. Pinpoint the key factors to distinguish these two responses.
2. Conclude your comparison by providing a final decision on which response is better, or they are tied. Begin your final decision statement with "So, the final decision is Response 1 / Response 2 / Tie". Ensure that your decision aligns coherently with the comprehensive evaluation and comparison you've provided.

Table 17: Input format of AUTO-J for pairwise response comparison protocol.

Write critiques for a submitted response on a given user's query, and grade the response:

[BEGIN DATA]
***
[Query]: prompt
***
[Response]: response
***
[END DATA]

Write critiques for this response. After that, you should give a final rating for the response on a scale of 1 to 10 by strictly following this format: "[[rating]]", for example: "Rating: [[5]]".

Table 18: Input format of AUTO-J for single-response evaluation protocol.

**Template for the unified format**

1. The key factors to distinguish these two responses:
   - (factor 1): (explanation 1)
   - (factor 2): (explanation 2)
...
2. The final decision:
   So, the final decision is Response 1 / Response 2 / Tie. (a few summarizing remarks)

**An instantiated example**

1. The key factors to distinguish these two responses:
   - pointing out: Both responses point out the changes made to the script and explain how these changes will solve the user's problem. However, Response 2 provides a more precise explanation of how the "-NoInteractive" flag will prevent any prompts during the execution of the script.
   - accuracy of the solution: Response 2's solution is more accurate as the "-NoInteractive" flag is the correct way to prevent any prompts or interactive input during the execution of a PowerShell script.
   - instruction following: Both responses follow the user's instructions to provide a solution that allows the script to run without waiting for user input. However, Response 2's solution is more in line with the user's need to run the script automatically.
   - coverage: Both responses provide a solution to the user's problem, but Response 2's solution is more accurate and directly addresses the user's need to run the script automatically without user input.
   - relevance to the user's problem: Response 2's solution is more relevant to the user's problem as it directly addresses the need to run the script automatically without user input.
   - code correctness: Both responses provide correct code, but Response 2's solution is more appropriate for the user's problem.
2. The final decision:
   So, the final decision is Response 2. Although both responses provide valid solutions, Response 2's solution is more accurate and directly addresses the user's need to run the script automatically without user input.

Table 19: The unified judgment format for pairwise response comparison training data, as well as a specific example for the "code_correction_rewriting" scenario.

**Query**

give me a trip plan for 5 days in France

**Response**

Sure, here's a potential 5-day trip plan for France:

Day 1: Arrival in Paris and Exploring the City

* Arrive at Charles de Gaulle airport and take a taxi or public transportation to your hotel in central Paris
* Check into your hotel and freshen up
* Visit the Eiffel Tower, one of the most iconic landmarks in the world
* Walk along the Champs-Élysées, a famous avenue known for its luxury shops and restaurants
* Explore the Louvre, one of the world's most famous museums, and see some of the world's most famous artworks
* Dinner at a classic Parisian bistro

Day 2: Day Trip to Versailles

* Take a high-speed train from Paris to Versailles
* Visit the Palace of Versailles, a magnificent palace and gardens built by the French monarchy
* Explore the Palace's grand halls and grand staircases
* Walk through the beautiful gardens and have a picnic
* Return to Paris in the evening

Day 3: Exploring Montmartre and the Artists' Quarter

* Take a metro to Montmartre, a charming neighborhood known for its artistic history
* Visit the Basilique du Sacré-Cœur, a beautiful church with stunning views of Paris
* Explore the streets and alleys of Montmartre and visit the works of famous artists like Van Gogh, Dali, and Picasso
* Have lunch at a traditional bistro
* Visit the Musée d'Orsay, a museum with an impressive collection of French art

Day 4: Trip to the Côte d'Azur

* Take a high-speed train from Paris to the Côte d'Azur, also known as the French Riviera
* Visit the Monaco Grand Prix circuit and see where the famous Formula One race takes place
* Walk along the Promenade des Anglais, a famous seaside promenade
* Have lunch at a seaside restaurant
* Check into your hotel in the Côte d'Azur

**Critiques from GPT-4 without scenario criteria as the system message**

The response, while detailed and comprehensive, does not take into account the feasibility of the proposed itinerary. On Day 4, the plan suggests a trip from Paris to the Côte d'Azur, which is a significant distance and would consume a considerable amount of time. This could potentially leave the traveler with little time to enjoy the activities listed for the day. Furthermore, the response does not provide any suggestions for Day 5 of the trip, leaving the user without a complete 5-day plan as requested in the query.

**Critiques from GPT-4 with scenario criteria as the system message**

The response does not provide a plan for the fifth day of the trip, which is a significant omission given the user's request for a 5-day trip plan. This makes the plan incomplete and not fully aligned with the user's query.
While the plan is generally well-structured, it lacks a clear timeline for each day's activities. Providing specific timeframes for each activity would make the plan more practical and easier to follow.
The plan also does not consider potential constraints such as the feasibility of visiting multiple major attractions in a single day, which may not be realistic given factors like travel time, waiting lines, and the time needed to fully appreciate each site.
The plan could also be more creative and personalized. For example, it could suggest less touristy, off-the-beaten-path attractions or activities that could provide a unique and memorable experience.
Lastly, the plan does not provide any information on the budget, which is an essential aspect of trip planning. Including estimated costs for transportation, accommodation, meals, and entrance fees would make the plan more thorough and useful.

**The combined final judgment**

The response provides a detailed and comprehensive itinerary for a trip to France, however, it falls short in several areas. Firstly, the plan does not fully align with the user's query as it only provides an itinerary for four days, leaving the fifth day unaccounted for. This is a significant omission given the user's specific request for a 5-day trip plan.
Secondly, the feasibility of the proposed itinerary is questionable. For instance, the plan suggests a trip from Paris to the Côte d'Azur on Day 4, which is a considerable distance and would consume a significant amount of time. This could potentially leave the traveler with little time to enjoy the activities listed for the day.
The plan also does not consider potential constraints such as the feasibility of visiting multiple major attractions in a single day. Factors like travel time, waiting lines, and the time needed to fully appreciate each site have not been taken into account. Providing specific timeframes for each activity would make the plan more practical and easier to follow.
Furthermore, the plan lacks creativity and personalization. It could be improved by suggesting less touristy, off-the-beaten-path attractions or activities that could provide a unique and memorable experience.
Lastly, the plan does not provide any information on the budget, which is an essential aspect of trip planning. Including estimated costs for transportation, accommodation, meals, and entrance fees would make the plan more thorough and useful.
In conclusion, while the response is well-structured and detailed, it lacks completeness, feasibility, and personalization. Therefore, the response could be significantly improved by addressing these issues.
Rating: [[5]]

Table 20: An example for collecting a complete evaluation judgment for single-response evaluation. We show the query, response, raw GPT-4 critiques with and without scenario criteria as the system message, and the final judgment by combining the previous two critiques.

| Label Distribution (Label, # of Samples) | | | | | |
|---|---|---|---|---|---|
| Win | 1594 | Lose | 1596 | Tie | 246 |

| Source Dataset Distribution (Source, # of Samples) | | | | | |
|---|---|---|---|---|---|
| Chatbot Arena Conversations | 2801 | OpenAI Summary | 100 | OpenAI WebGPT | 45 |
| PKU-SafeRLHF | 158 | Stanford SHP | 81 | Synthetic GPT-J | 251 |

| Scenario Distribution (Name, # of Samples) | | | | | |
|---|---|---|---|---|---|
| ranking | 100 | open_question | 100 | text_correction | 18 |
| recommendation | 100 | post_summarization | 100 | writing_product_description | 16 |
| creative_writing | 100 | writing_song_lyrics | 98 | language_polishing | 15 |
| planning | 100 | functional_writing | 94 | code_to_code_translation | 15 |
| brainstorming | 100 | writing_cooking_recipe | 88 | writing_legal_document | 13 |
| exam_question_without_math | 100 | code_correction_rewriting | 86 | writing_blog_post | 13 |
| roleplay | 100 | writing_personal_essay | 84 | title_generation | 12 |
| text_summarization | 100 | analyzing_general | 67 | writing_social_media_post | 12 |
| asking_how_to_question | 100 | explaining_code | 59 | reading_comprehension | 11 |
| chitchat | 100 | information_extraction | 51 | writing_technical_document | 10 |
| verifying_fact | 100 | writing_email | 51 | text_simplification | 10 |
| value_judgment | 100 | writing_job_application | 46 | keywords_extraction | 6 |
| code_generation | 100 | classification_identification | 44 | writing_scientific_paper | 5 |
| text_to_text_translation | 100 | writing_presentation_script | 42 | writing_marketing_materials | 4 |
| math_reasoning | 100 | exam_question_with_math | 41 | topic_modeling | 3 |
| question_generation | 100 | data_analysis | 39 | writing_news_article | 3 |
| counterfactual | 100 | instructional_rewriting | 30 | note_summarization | 2 |
| seeking_advice | 100 | paraphrasing | 27 | code_simplification | 1 |
| explaining_general | 100 | writing_advertisement | 20 | others | 100 |

Table 21: Statistics for pairwise training data: the distribution of labels, source datasets, and scenarios.

| Score Distribution (Score, # of Samples) | | | | | |
|---|---|---|---|---|---|
| 1 | 29 | 2 | 137 | 3 | 178 |
| 4 | 210 | 5 (5.5) | 131 | 6 (6.5) | 241 |
| 7 | 27 | 8 | 4 | 10 | 3 |

| Scenario Distribution (Name, # of Samples) | | | | | |
|---|---|---|---|---|---|
| code_generation | 24 | explaining_code | 18 | writing_technical_document | 15 |
| explaining_general | 23 | functional_writing | 18 | text_simplification | 15 |
| open_question | 23 | writing_song_lyrics | 18 | language_polishing | 15 |
| seeking_advice | 23 | ranking | 18 | code_to_code_translation | 15 |
| math_reasoning | 22 | planning | 17 | writing_blog_post | 15 |
| chitchat | 21 | classification_identification | 17 | reading_comprehension | 14 |
| value_judgment | 21 | exam_question_with_math | 17 | topic_modeling | 14 |
| brainstorming | 21 | writing_cooking_recipe | 17 | writing_advertisement | 14 |
| creative_writing | 20 | writing_email | 17 | title_generation | 14 |
| roleplay | 20 | information_extraction | 17 | keywords_extraction | 14 |
| verifying_fact | 20 | paraphrasing | 17 | writing_legal_document | 14 |
| counterfactual | 19 | code_correction_rewriting | 17 | writing_news_article | 14 |
| asking_how_to_question | 19 | data_analysis | 16 | writing_social_media_post | 14 |
| exam_question_without_math | 19 | writing_product_description | 16 | code_simplification | 12 |
| text_summarization | 19 | instructional_rewriting | 16 | writing_scientific_paper | 12 |
| recommendation | 18 | writing_presentation_script | 16 | writing_marketing_materials | 8 |
| question_generation | 18 | analyzing_general | 16 | note_summarization | 4 |
| text_to_text_translation | 18 | writing_job_application | 16 | writing_biography | 4 |
| writing_personal_essay | 18 | text_correction | 16 | others | 27 |

Table 22: Statistics for single training data: the distribution of GPT-4 ratings, and scenarios.

| | GPT-4 judgments | | | | | | Human judgments | | | | | |
|---|---|---|---|---|---|---|---|---|---|---|---|---|
| Baseline | SelFee | | | Vicuna | | | SelFee | | | Vicuna | | |
| judgment | Win | Tie | Lose | Win | Tie | Lose | Win | Tie | Lose | Win | Tie | Lose |
| Summarization | 12 | 0 | 0 | 9 | 0 | 3 | 10 | 1 | 1 | 11 | 1 | 0 |
| Exam Questions | 11 | 1 | 0 | 7 | 1 | 4 | 8 | 2 | 2 | 7 | 2 | 3 |
| Code | 20 | 0 | 0 | 15 | 0 | 5 | 15 | 2 | 3 | 11 | 5 | 4 |
| Rewriting | 18 | 0 | 2 | 18 | 0 | 2 | 14 | 5 | 1 | 14 | 4 | 2 |
| Creative Writing | 33 | 0 | 3 | 29 | 0 | 7 | 26 | 7 | 3 | 24 | 9 | 3 |
| Functional Writing | 37 | 0 | 3 | 28 | 0 | 12 | 37 | 2 | 1 | 32 | 6 | 2 |
| General Communication | 47 | 0 | 1 | 39 | 0 | 9 | 43 | 2 | 3 | 39 | 6 | 3 |
| NLP Tasks | 40 | 0 | 4 | 35 | 0 | 9 | 29 | 7 | 8 | 22 | 15 | 7 |
| Overall | 218 | 1 | 13 | 180 | 1 | 51 | 182 | 28 | 22 | 160 | 48 | 24 |
| Baseline | L2Chat | | | ChatGPT | | | L2Chat | | | ChatGPT | | |
| judgment | Win | Tie | Lose | Win | Tie | Lose | Win | Tie | Lose | Win | Tie | Lose |
| Summarization | 10 | 0 | 2 | 12 | 0 | 0 | 10 | 1 | 1 | 10 | 1 | 1 |
| Exam Questions | 11 | 0 | 1 | 10 | 0 | 2 | 6 | 3 | 3 | 6 | 1 | 5 |
| Code | 18 | 0 | 2 | 16 | 0 | 4 | 11 | 5 | 4 | 8 | 6 | 6 |
| Rewriting | 17 | 1 | 2 | 14 | 0 | 6 | 10 | 7 | 3 | 9 | 8 | 3 |
| Creative Writing | 30 | 0 | 6 | 26 | 2 | 8 | 13 | 14 | 9 | 16 | 15 | 5 |
| Functional Writing | 32 | 0 | 8 | 22 | 2 | 16 | 23 | 13 | 4 | 23 | 14 | 3 |
| General Communication | 41 | 0 | 7 | 36 | 0 | 12 | 28 | 15 | 5 | 29 | 10 | 9 |
| NLP Tasks | 37 | 1 | 6 | 35 | 1 | 8 | 13 | 22 | 9 | 16 | 17 | 11 |
| Overall | 196 | 2 | 34 | 171 | 5 | 56 | 114 | 80 | 38 | 117 | 72 | 43 |
| Baseline | WizardLM | | | GPT-4 | | | WizardLM | | | GPT-4 | | |
| judgment | Win | Tie | Lose | Win | Tie | Lose | Win | Tie | Lose | Win | Tie | Lose |
| Summarization | 10 | 0 | 2 | 8 | 0 | 4 | 11 | 1 | 0 | 6 | 2 | 4 |
| Exam Questions | 11 | 1 | 0 | 3 | 0 | 9 | 6 | 3 | 3 | 2 | 2 | 8 |
| Code | 17 | 0 | 3 | 12 | 0 | 8 | 9 | 7 | 4 | 6 | 5 | 9 |
| Rewriting | 15 | 1 | 4 | 12 | 0 | 8 | 12 | 6 | 2 | 12 | 3 | 5 |
| Creative Writing | 31 | 1 | 4 | 23 | 0 | 13 | 17 | 17 | 2 | 11 | 9 | 16 |
| Functional Writing | 30 | 1 | 9 | 17 | 1 | 22 | 24 | 12 | 4 | 27 | 6 | 7 |
| General Communication | 38 | 1 | 9 | 28 | 0 | 20 | 32 | 14 | 2 | 35 | 3 | 10 |
| NLP Tasks | 35 | 2 | 7 | 23 | 1 | 20 | 19 | 14 | 11 | 8 | 11 | 25 |
| Overall | 187 | 7 | 38 | 126 | 2 | 104 | 130 | 74 | 28 | 107 | 41 | 84 |

Table 23: Detailed comparison results between critiques generated by AUTO-J and baselines for single-response evaluation. Results on left side are GPT-4 judgments, and results on right side are human judgments. Vicuna, L2Chat, and WizardLM respectively stand for Vicuna-13B-v1.5, LLaMA-2-Chat-13B, and WizardLM-13B-v1.2.

| Model | Auto-J Rating | GPT-4 Win-rate | Rank | | |
| --- | --- | --- | --- | --- | --- |
| | | | GPT-4 | Auto-J | Δ |
| XwinLM 70b V0.1 | 5.694 | 95.57 | 1 | 1 | 0 |
| LLaMA2 Chat 70B | 5.678 | 92.66 | 2 | 2 | 0 |
| XwinLM 13b V0.1 | 5.647 | 91.76 | 3 | 3 | 0 |
| OpenChat V3.1 13B | 5.532 | 89.49 | 4 | 8 | 4 |
| WizardLM 13B V1.2 | 5.547 | 89.17 | 5 | 6 | 1 |
| Vicuna 33B v1.3 | 5.570 | 88.99 | 6 | 5 | -1 |
| Humpback LLaMa2 70B | 5.498 | 87.94 | 7 | 11 | 4 |
| XwinLM 7b V0.1 | 5.584 | 87.83 | 8 | 4 | -4 |
| OpenBudddy-LLaMA2-70B-v10.1 | 5.448 | 87.67 | 9 | 14 | 5 |
| OpenChat V2-W 13B | 5.533 | 87.13 | 10 | 7 | -3 |
| OpenBuddy-LLaMA-65B-v8 | 5.458 | 86.53 | 11 | 13 | 2 |
| WizardLM 13B V1.1 | 5.497 | 86.32 | 12 | 12 | 0 |
| OpenChat V2 13B | 5.519 | 84.97 | 13 | 9 | -4 |
| Humpback LLaMa 65B | 5.379 | 83.71 | 14 | 19 | 5 |
| Vicuna 13B v1.3 | 5.388 | 82.11 | 15 | 18 | 3 |
| OpenBuddy-LLaMA-30B-v7.1 | 5.391 | 81.55 | 16 | 17 | 1 |
| LLaMA2 Chat 13B | 5.518 | 81.09 | 17 | 10 | -7 |
| OpenChat-13B | 5.437 | 80.87 | 18 | 15 | -3 |
| OpenBuddy-Falcon-40B-v9 | 5.373 | 80.70 | 19 | 20 | 1 |
| UltraLM 13B | 5.342 | 80.64 | 20 | 22 | 2 |
| OpenChat8192-13B | 5.429 | 79.54 | 21 | 16 | -5 |
| OpenCoderPlus-15B | 5.357 | 78.70 | 22 | 21 | -1 |
| OpenBudddy-LLaMA2-13B-v11.1 | 5.340 | 77.49 | 23 | 23 | 0 |
| Vicuna 7B v1.3 | 5.332 | 76.84 | 24 | 25 | 1 |
| WizardLM 13B | 5.247 | 75.31 | 25 | 32 | 7 |
| JinaChat | 5.319 | 74.13 | 26 | 26 | 0 |
| airoboros 65B | 5.318 | 73.91 | 27 | 27 | 0 |
| airoboros 33B | 5.289 | 73.29 | 28 | 30 | 2 |
| Guanaco 65B | 5.313 | 71.80 | 29 | 29 | 0 |
| LLaMA2 Chat 7B | 5.334 | 71.37 | 30 | 24 | -6 |
| Vicuna 13B | 5.314 | 70.43 | 31 | 28 | -3 |
| OpenBuddy-Falcon-7b-v6 | 5.214 | 70.36 | 32 | 34 | 2 |
| Baize-v2 13B | 5.165 | 66.96 | 33 | 38 | 5 |
| LLaMA 33B OASST RLHF | 5.173 | 66.52 | 34 | 37 | 3 |
| Minotaur 13B | 5.210 | 66.02 | 35 | 36 | 1 |
| Guanaco 33B | 5.212 | 65.96 | 36 | 35 | -1 |
| Nous Hermes 13B | 5.271 | 65.47 | 37 | 31 | -6 |
| Vicuna 7B | 5.237 | 64.41 | 38 | 33 | -5 |
| Baize-v2 7B | 5.083 | 63.85 | 39 | 39 | 0 |
| LLaMA 33B OASST SFT | 4.985 | 54.97 | 40 | 41 | 1 |
| Guanaco 13B | 5.027 | 52.61 | 41 | 40 | -1 |
| ChatGLM2-6B | 4.846 | 47.13 | 42 | 46 | 4 |
| Guanaco 7B | 4.943 | 46.58 | 43 | 43 | 0 |
| Falcon 40B Instruct | 4.934 | 45.71 | 44 | 44 | 0 |
| Alpaca Farm PPO Sim (GPT-4) 7B | 4.978 | 44.10 | 45 | 42 | -3 |
| Pythia 12B SFT | 4.809 | 41.86 | 46 | 47 | 1 |
| Alpaca Farm PPO Human 7B | 4.907 | 41.24 | 47 | 45 | -2 |
| Cohere Chat | 4.524 | 29.57 | 48 | 51 | 3 |
| Cohere | 4.522 | 28.39 | 49 | 52 | 3 |
| Alpaca 7B | 4.658 | 26.46 | 50 | 48 | -2 |
| Pythia 12B OASST SFT | 4.620 | 25.96 | 51 | 49 | -2 |
| Falcon 7B Instruct | 4.537 | 23.60 | 52 | 50 | -2 |
| Baichuan-13B-Chat | 4.291 | 21.80 | 53 | 53 | 0 |

Table 24: Values and ranking by Auto-J and GPT-4 for open-source LLMs on AlpacaEval. Value of AUTO-J is the model's average rating on AlpacaEval dataset assigned by AUTO-J in single-response evaluation protocol, and value of GPT-4 is the model's win-rate against Davinci003 determined by GPT-4 on AlpacaEval dataset. $\Delta = \text{Rank}_{\text{Auto-J}} - \text{Rank}_{\text{GPT-4}}$

**Query**

Draft an email to my family telling them I booked flights for Thanksgiving. I'll arrive on November 22 and leave on the 30th

| Response 1 (from GPT4ALL-13B-snoozy) | Response 2 (Preferred, from Claude-v1) |
|---|---|
| Subject: Exciting News: Flights Booked for Thanksgiving!
Dear Family,
I hope this email finds you all doing well! I am thrilled to share some exciting news with you all. I have booked my flights for Thanksgiving! I will be arriving on November 22 and leaving on November 30.
I cannot wait to spend quality time with all of you and enjoy the delicious Thanksgiving feast together. There might be a few surprises in store for you all, so make sure to keep an open mind!
I have attached the flight itinerary for your reference. Please feel free to reach out to me if you have any questions or concerns.
Wishing you all a very happy Thanksgiving!
Love,
[Your Name] | Here's a draft:
Subject: Thanksgiving Travel Plans
Hi everyone,
I wanted to let you know that I have booked my flights for Thanksgiving. I will be arriving on Wednesday, November 22nd and departing on Thursday, November 30th. I'm looking forward to seeing you all and spending the holiday together.
Please let me know if there's anything I should bring or if there are any plans already in the works for meals or activities. I'm happy to contribute however I can.
Safe travels to anyone else coming into town. See you all very soon!
Love,
[Your name]
Feel free to modify the email as you see fit. Just let me know if you'd like me to make any changes to the email before you send it. |

**Judgment by Auto-J (✓)**

1. The key factors to distinguish these two responses:
  - purpose: Both responses meet this criterion as they both inform the family about the booked flights for Thanksgiving. However, Response 2 does a better job by asking for any plans for meals or activities, which aligns more closely with the purpose of the email, which is to inform and also to plan for the Thanksgiving celebration.
  - tone: Both responses maintain a casual and friendly tone, which is appropriate for the situation. However, Response 1's tone is more formal and serious, while Response 2's tone is more casual and friendly, which is more suitable for a family email.
  - body: Both responses provide details about the trip in the body of the email. However, Response 2 provides more detailed information about the trip, such as the arrival and departure dates, and also asks for any plans for meals or activities, which makes it more informative and engaging.
  - personalization: Response 2 is more personalized as it asks for any plans for meals or activities, which shows that the sender is interested in the family's plans and wants to be involved in the celebration.
  - information richness: Response 2 provides more detailed information about the trip and also asks for any plans for meals or activities, which makes it richer in information compared to Response 1.
2. The final decision:
  So, the final decision is Response 2. Although both responses meet the criteria and are well-written, Response 2 is better because it is more informative, engaging, and personalized, which makes it more suitable for a family email.

**Judgment by ChatGPT (✗)**

Assistant A's response is more detailed and provides a warm and enthusiastic tone. The email includes a subject line that captures attention and clearly states the purpose of the email. It also mentions surprises and encourages the family to keep an open mind, adding a touch of excitement. The email includes an attached flight itinerary for reference and invites the family to reach out with any questions or concerns. Overall, Assistant A's response is helpful, relevant, accurate, and provides a good level of detail.
Assistant B's response is also helpful and relevant. It includes a subject line that states the purpose of the email and provides the necessary information about the dates of arrival and departure. It also offers to contribute and asks if there are any plans or items to bring. However, it lacks the same level of enthusiasm and detail as Assistant A's response. It does not mention surprises or provide an attached flight itinerary. While it is a straightforward and concise email, it could benefit from more personalization and excitement.
Based on the factors mentioned above, Assistant A provides a better response. [[A]]

**Judgment by LLaMA-2-Chat-13B (✗)**

Sure, I'd be happy to help! Here's my evaluation of the two responses:
Both Assistant A and Assistant B provide helpful and relevant responses to the user's question. However, there are some differences in the quality of their responses.
Assistant A's response is more detailed and creative. The email draft provided is well-written and includes a personal touch with the mention of potential surprises. The language used is also more formal and polished, which is appropriate for a family email. Additionally, Assistant A provides a flight itinerary for reference, which is a thoughtful touch.
Assistant B's response, on the other hand, is more straightforward and to the point. The email draft provided is simple and easy to understand, but it lacks the personal touch and creativity of Assistant A's response. Assistant B also uses a more casual tone, which may not be appropriate for a family email.
Based on these factors, I would say that Assistant A provides a better response to the user's question. Assistant A's response is more detailed, creative, and formal, while Assistant B's response is more straightforward and casual. Therefore, my final verdict is: [[A]]

Table 25: The complete judgments given by different models on a pairwise response comparison example. We mark if each judgment agrees with human preference as well, where ✓ stands for agreement and ✗ stands for disagreement. (Part 1)

| **Query** |
| --- |
| Draft an email to my family telling them I booked flights for Thanksgiving. I'll arrive on November 22 and leave on the 30th |

| **Response 1** (from GPT4ALL-13B-snoozy) | **Response 2** (Preferred, from Claude-v1) |
| --- | --- |
| Subject: Exciting News: Flights Booked for Thanksgiving! Dear Family, I hope this email finds you all doing well! I am thrilled to share some exciting news with you all. I have booked my flights for Thanksgiving! I will be arriving on November 22 and leaving on November 30. I cannot wait to spend quality time with all of you and enjoy the delicious Thanksgiving feast together. There might be a few surprises in store for you all, so make sure to keep an open mind! I have attached the flight itinerary for your reference. Please feel free to reach out to me if you have any questions or concerns. Wishing you all a very happy Thanksgiving! Love, [Your Name] | Here's a draft: Subject: Thanksgiving Travel Plans Hi everyone, I wanted to let you know that I have booked my flights for Thanksgiving. I will be arriving on Wednesday, November 22nd and departing on Thursday, November 30th. I'm looking forward to seeing you all and spending the holiday together. Please let me know if there's anything I should bring or if there are any plans already in the works for meals or activities. I'm happy to contribute however I can. Safe travels to anyone else coming into town. See you all very soon! Love, [Your name] Feel free to modify the email as you see fit. Just let me know if you'd like me to make any changes to the email before you send it. |

| **Judgment by PandaLM (✗)** |
| --- |
| 1 ### Reason: Response 1 is more concise and follows the instruction more closely. ### Reference: Subject: Flights Booked for Thanksgiving! Dear Family, I hope this email finds you all doing well! I am thrilled to share some exciting news with you all. I have booked my flights for Thanksgiving! I will be arriving on November 22 and leaving on November 30. I cannot wait to spend quality time with all of you and enjoy the delicious Thanksgiving feast together. There might be a few surprises in store for you all, so make sure to keep an open mind! I have attached the flight itinerary for your reference. Please feel free to reach out to me if you have any questions or concerns. Wishing you all a very happy Thanksgiving! Love, [Your Name] |

| **Judgment by Vicuna-13B-v1.5 (✗)** |
| --- |
| After comparing the two responses, I find that Assistant A's response is more helpful, relevant, accurate, and creative. The response is well-structured, with a clear subject line and a warm greeting. It provides the necessary information about the travel plans and expresses excitement about spending time with the family. The response also includes a touch of creativity with the mention of surprises. Assistant B's response is also good, but it lacks the depth and creativity of Assistant A's response. The response is more straightforward and lacks a personal touch. It does ask for input on meals and activities, which is a good point, but it could have been more engaging. Based on these factors, I conclude that Assistant A's response is better, and the final verdict is "[[A]]". |

| **Judgment by WizarLM-13B-v1.2 (✗)** |
| --- |
| After evaluating both responses, I would rate Assistant A as the better one. Here's why: Assistant A's response is more concise and directly answers the user's question. It provides all the necessary information, including the dates of arrival and departure, and a brief message expressing excitement to spend Thanksgiving with the family. The email is well-structured and professional, with a clear subject line and a warm, personalized closing. Assistant B's response, while polite and friendly, is less focused on the main request. It asks for input on meals and activities, which is not directly related to the user's question. The response also seems to be more of a draft rather than a final email, as it invites the user to modify the email before sending it. Therefore, based on the factors of helpfulness, relevance, accuracy, depth, creativity, and level of detail, I would rate Assistant A as the better response. |

| **Judgment by GPT-4 (✓)** |
| --- |
| Both assistants provided good responses, drafting an email that communicates the user's travel plans for Thanksgiving. However, Assistant B's response is slightly more interactive and considerate, asking if there's anything they should bring or if there are any plans already in the works for meals or activities. This shows a willingness to contribute to the family gathering. Assistant B also acknowledges that other family members might be traveling, wishing them safe travels. Therefore, Assistant B's response is more comprehensive and thoughtful. Final Verdict: [[B]] |

Table 26: The complete judgments given by different models on a pairwise response comparison example. We mark if each judgment agrees with human preference as well, where ✓ stands for agreement and ✗ stands for disagreement. (Part 2)

| Query |
| --- |
| What's the best way to cook salmon on the grill. |

| Response |
| --- |
| The best way to cook salmon on the grill is to preheat the grill to medium-high heat, brush the salmon with olive oil, season with salt and pepper, and place skin-side down on the grill. Allow the salmon to cook for 5-7 minutes, until the skin is crispy and the fish flakes easily with a fork. Flip the salmon, and cook for an additional 3-4 minutes. Serve with your favorite sides and enjoy. |

| Critique from AUTO-J |
| --- |
| The response provided is generally informative and offers a basic method for grilling salmon. However, it lacks depth and detail in several key areas. A main shortcoming is the absence of specific temperatures or cooking times, which are crucial for achieving the perfect grilled salmon. The response vaguely states "medium-high heat" and "5-7 minutes" for the first side of cooking, and "3-4 minutes" for the second side. More precise guidance would be beneficial for users who may not be experienced grill masters. |
| The response also fails to address the quality of the salmon that is required for the recipe. It is essential to mention the type of salmon that would be best suited for this method of cooking. Furthermore, the response does not provide detailed information on how to determine when the salmon is cooked. This could potentially lead to overcooked or undercooked salmon, which is a common problem for novice cooks. |
| The response could have been more interactive and personalized by asking the user about their specific grill or their experience with cooking salmon. This would have made the response more engaging and helpful. Additionally, the response could have been more professional by citing credible sources or providing references to back up the cooking instructions. This would have added credibility to the response. |
| Lastly, the response missed an opportunity to provide a comprehensive answer by not addressing potential questions that the user might have, such as what to do if the salmon sticks to the grill or how to ensure the salmon is cooked evenly. |
| In conclusion, while the response is a good starting point, it lacks the necessary detail and personal touch that would make it a truly excellent response. |

| Critique from GPT-4 (AUTO-J wins) |
| --- |
| The response is quite comprehensive and provides a clear, step-by-step guide on how to grill salmon. It includes details such as the heat level of the grill, how to prepare the salmon, how long to cook it on each side, and how to determine when it's done. However, it could be improved by including some tips on choosing the best piece of salmon for grilling, or suggesting some specific side dishes that pair well with grilled salmon. |

| Critique from ChatGPT (AUTO-J wins) |
| --- |
| 1. Lack of details: The response could benefit from providing more specific instructions. For example, it does not mention the thickness of the salmon fillet or whether the grill should be covered or uncovered during cooking. |
| 2. Absence of alternative methods: While the provided method is a good starting point, it would be helpful to mention other techniques such as using a cedar plank or aluminum foil to cook the salmon. |
| 3. Limited seasoning options: While salt and pepper are mentioned as seasonings, it would be nice to suggest additional herbs or spices that could enhance the flavor of the salmon. |

| Critique from SelFee (AUTO-J wins) |
| --- |
| The answer is clear and concise, providing step-by-step instructions on how to cook salmon on the grill. However, it would be helpful to mention the ideal grill temperature and suggest using a grill brush to clean the grates before cooking. Additionally, it would be useful to suggest using a thermometer to check the internal temperature of the salmon to ensure it is cooked to a safe temperature. Overall, I would give this answer a score of 8. Revision is needed. |

| Critique from LLaMA-2-Chat-13B (AUTO-J wins) |
| --- |
| 1. Lack of detail: The response could benefit from more detail on the cooking process. For example, the user might want to know the exact temperature of the grill, or how to tell when the salmon is done. |
| 2. Limited seasoning options: The response only mentions salt and pepper as seasoning options, but there are many other delicious seasonings that could be used to enhance the flavor of the salmon. |
| 3. No visuals: The response does not include any visuals or images to help the user understand the cooking process or the final product. |

| Critique from Vicuna-13B-v1.5 (AUTO-J wins) |
| --- |
| Critique 1: The response provides a clear and concise method for cooking salmon on the grill. It includes important details such as preheating the grill and brushing the salmon with olive oil. However, it would be helpful to include more information about the cooking time and temperature, as well as tips for checking the doneness of the salmon. |
| Critique 2: The response is well-written and easy to follow, with clear instructions for seasoning and cooking the salmon. It also suggests serving the salmon with favorite sides, which adds a personal touch. However, the response could benefit from more variety in the language and a few more details about the cooking process, such as the importance of flipping the salmon and monitoring the grill temperature. |

| Critique from Wizardlm-13B-v1.2 (Tie) |
| --- |
| 1. The response is concise and provides a clear, step-by-step guide on how to cook salmon on the grill. It mentions the preheating of the grill, seasoning the salmon, and the cooking time, which are all essential details. |
| 2. The response could benefit from more detail on the type of olive oil to use and the seasoning options. For example, extra virgin olive oil is a good choice for grilling, and other seasoning options could include lemon juice, garlic powder, or smoked paprika. |
| 3. The response does not mention the importance of oiling the grates before cooking, which can prevent the salmon from sticking to the grill. |
| 4. The response does not provide any information on how to tell if the salmon is cooked properly. It would be helpful to mention that the salmon should flake easily with a fork and have a slight pink color in the center. |
| 5. The response does not suggest any sides to serve with the salmon, which could be a helpful addition for users who are unsure of what to serve with grilled salmon. |

Table 27: Critiques given by different models for a response. We remove the "rating" part in each critique, and mark the comparison results between AUTO-J and baselines judged by human.

**Query**

My Ex, to whom I was never married, just served me with divorce papers TLDR - I'm in the process of evicting my ex from my house in Kansas. He just served me with divorce papers, though we were never married. He's claiming it was "common law" and I think he's trying to get my house. This is just so strange that I feel like I'm living in another universe. I hope I can get a little advice. This is all happening in Kansas. Anyway, my ex and I were together for about 10 years. Almost 8 years ago I bought a house. I bought the house. Me. Alone. My down payment, my credit, my mortgage. He didn't pay anything. His name was never on anything. Not the title, not the insurance, not the mortgage, not even the utilities. We did have an unwritten agreement that he'd pay half the mortgage, utilities, and other shared expenses like groceries. And for about a year, maybe, he did pay me varying amounts of money. It was never half, and it caused a lot of arguments. Whatever he paid me, he paid in cash. Since summer 2011 I have paid all of it. He pays for whatever food he eats outside of groceries and covers whatever expenses he has. I don't know anything about that. He runs "businesses" that include a youtube channel, a Twitch channel, and he's in a band. I don't have any idea how much money he made at them, or if he made any at all. For a number of reasons, including this ridiculous financial situation, I ended our relationship in mid-March. I told him I was planning to move out of state without him and that he needed to leave the house. I told him to be out by mid July. He totally ignored me and did nothing. July came and went. I moved out of state. He stayed in my house. I began communicating with him only by email - I told him I wanted to have the house on the market by October, that he had to be out etc. He continued to ignore me, and I had to get a lawyer who served him with a notice of eviction at the beginning of October. He challenged the eviction and the court date is the first week of November. Through all of this the ex refused to communicate. I never had any idea what he was thinking. Then yesterday I was served with divorce papers. What I have has no details, but a friend of ours told me that the ex is saying he's entitled to the house (or some part of) because it is community property from our common law marriage. Is this insane? Because it feels insane. I know I have to deal with it, and I will, but what the heck? We never discussed being married. I never told anyone we were married, and I never heard of him doing so. We never used each other's last names. I have always filed my taxes as single. Does he have any hope here at all? Is it possible to accidentally get married? Without consent and against my will? I was REALLY hoping to be done with this nonsense. Thanks to you all for whatever you can tell me.

**Response with the highest probability among 32 candidates (AUTO-J: 4.0, GPT-4: 4.0)**

It is not common law marriage, and Kansas does not recognize common law marriage. Living together for a certain period of time does not automatically make a couple married, even if they believe they are married. Common law marriage requires a valid marriage ceremony and a mutual agreement to be married. It is important to consult with an attorney to understand the specific details of your situation and the laws in your state.

**Best-of-8 and 16 response selected by AUTO-J (AUTO-J: 5.0, GPT-4: 6.0)**

It sounds like you are going through a difficult and confusing situation. Based on the information you have provided, it seems unlikely that your ex could claim common law marriage or any other kind of property rights in the house, as there is no evidence that you two were ever married or that you intended to be married.26emHowever, it is possible that your ex could make a claim for property or assets that you acquired during your relationship, even if you were not married. Kansas is a community property state, which means that any property or assets acquired during a marriage are considered community property and are owned equally by both spouses. If your ex can prove that you acquired the house during your relationship, they may be able to make a claim to a portion of the property.

It is important to consult with a qualified attorney who can advise you on your specific legal situation and help you navigate the eviction and divorce process. They can review the divorce papers you have received and help you understand your rights and options. It may also be helpful to keep detailed records of all financial transactions and agreements between you and your ex to support your case in court.

**Best-of-32 response selected by AUTO-J (AUTO-J: 6.0, GPT-4: 9.0)**

It seems like you're experiencing a complex and bewildering situation. Based on the information you've provided, here's a summary of your situation and potential options:

   1. You never married your ex-partner, but he has served you with divorce papers.
   2. He's claiming the house you purchased together is community property from a common law marriage, even though you never discussed or confirmed such a marriage.
   3. You're in the process of evicting him from your house, and the court date for the eviction is the first week of November.
   4. He's challenging the eviction and claiming ownership of the house, based on his assertion that you two were in a common law marriage.
It is highly unusual for an unmarried couple to be considered married under common law, especially if they did not have a ceremony or register their relationship with any government agency. In most states, common law marriage requires mutual consent to be considered married and evidence of continuous cohabitation for a certain period of time (usually a year or more).

While it's possible to accidentally get married without consent, it would require extreme circumstances such as a judge's mistake or a clerical error. Typically, a couple must go through a formal ceremony, obtain a marriage license, and fulfill the requirements and paperwork for marriage.

Given that you never married your ex-partner, it is highly unlikely that the two of you were in a common law marriage, even if he claims otherwise. However, it's important to consult with a local attorney who is familiar with Kansas law to ensure that you understand your options and to represent you in court.

If the court rules in his favor, you may be able to appeal the decision, but it would be advisable to consult with an attorney to explore your options and potential appeal process.

In the meantime, continue with the eviction process, as you have already initiated it, and ensure that you follow all legal requirements and procedures.

Table 28: A Best-of-$N$ selection example to show the usefulness of AUTO-J's overall rating on single-response evaluation. The base LLM is Vicuna-7B-v1.5.

