# OpenReview forum: "Generative Judge for Evaluating Alignment"
_ICLR.cc/2024/Conference — ICLR 2024 poster_

### Official Review · Reviewer_PG92 · 2023-10-28

**Soundness:** 3 good
**Presentation:** 2 fair
**Contribution:** 3 good
**Rating:** 6
**Confidence:** 2

**Summary:**

The paper fine-tunes a language model to automatically judge the output of another language model, either evaluating a single generation or a pair of generations.

(Sorry that I do not think I understand the core training and evaluation setup of the paper; either it is my reading comprehension's problem or there might be issues with the presentation.)

**Strengths:**

The paper provides an open-sourced model that can automatically judge a models' generated output; this could potentially enable more researchers to run automatic evaluation at a lower cost with higher reliability.

**Weaknesses:**

- The papers is a large engineering effort (e.g., distilling GPT-4 for the task of evaluation) without much novel ideas (I do not think that a paper needs to be novel to be accepted, but this paper does score low in terms of novelty)
- The presentation of the method and contribution feels very confusing to me (maybe it's just my fault). See questions below. I do not know whether other reviewers would have similar concerns though.

**Questions:**

- If I understand correctly, did you define the scenarios and categories both for fine-tune the model to perform evaluation AND test the model to perform evaluation? In that case, if we want to evaluate an LM that performs a new task that is not in the 58 categories you have defined, would Auto-J generalize to categories unseen during training?
- Did you use GPT-4 as the label generator? If so, if one had enough OpenAI credit, the optimal strategy of evaluation according to this paper would still be using GPT-4 or not? (both yes/no are okay answers, but I think it'd be useful to clarify)

---

> ### Author Response · Authors · 2023-11-17
> **Response to Weakness 1**
>
> # Response to Weakness 1
>
> ---
>
> > “The papers is a large engineering effort (e.g., distilling GPT-4 for the task of evaluation) without much novel ideas (I do not think that a paper needs to be novel to be accepted, but this paper does score low in terms of novelty)”
>
> Thank you for your valuable feedback. We would like to use this opportunity to clarify the focus of our work and further detail the novel design in our data collection process.
>
> **The focus shift in LLM generation evaluation**: In this paper, we focus on the field of evaluating Large Language Models (LLMs) on how they align with human needs in massive real-world scenarios. This field has garnered increasing concern due to the growing capabilities of LLMs and their wider deployment across various scenarios.
>
> As our introduction notes, the field has been pivoting away from a narrow definition of novel design—such as designing specific algorithms, metrics, or model architectures for evaluators—toward a more user-centric evaluation approach. This shift reflects a collective recognition that the true measure of an LLM's utility is not solely captured by its performance under standard academic benchmarks but rather by its effectiveness in diverse, real-world contexts [1, 2, 3].
>
> **Our contribution under the practical necessity**: In response to the actual needs in this research field, we choose to pay attention more to the scenario definition, criteria design, and data generation under a more user-centric setting rather than developing elaborate new algorithms or model architectures. We show that the developed Auto-J has better evaluation capabilities than many competitive baselines with significantly more parameters (such as ChatGPT, Claude2, and LLaMA-2-70B-Chat). Since Auto-J achieves these results with only 13B parameters, it also lays a solid foundation for the research line of scalable oversight [4]: by combining human efforts with model capabilities, a small model can serve as a good overseer for much stronger models, ensuring effective oversight even if a model's abilities surpass human capabilities.
>
> **Novelty in data collection**: We want to emphasize that we do not rely solely on simple prompting GPT-4 for constructing the training data like previous works that also train a small judge model [5]. Instead, we employ a synergistic blend of GPT-4's advanced capabilities with meticulously crafted human inputs (such as scenario definitions and criteria design). The data compilation process is heavily reliant on human labor and ingenuity, encompassing finding the sources for user inquiries and potential model responses.
>
> We have also handcrafted a large collection of evaluation criteria, which effectively guide GPT-4 to generate more comprehensive and specific judgments than direct prompting without any guidance. We employ data filtering (using existing human labels) in pairwise comparison data collection and execute a “divide-and-conquer” strategy (combining judgments with and without reference to the criteria) in single response evaluation data collection to get more comprehensive and specific GPT-4 outputs for training.
>
> **Good performance with data efficiency**: With all the above-mentioned techniques, we have successfully developed Auto-J in a data-efficient manner with less than 5,000 training samples. We show with extensive experiments that Auto-J surpasses all baselines except GPT-4 in pairwise response comparison (Table 1) and even outperforms GPT-4 in critique generation (Figure 3).
>
> **Evaluator under low-resource settings**: Furthermore, although GPT-4 may perform better when resources (like time and budget) are ample, Auto-J is open-sourced, smaller, and faster. We assert that Auto-J should not be viewed as a “neither here nor there” substitute for GPT-4 but rather as a valuable complement for those with limited resources, large-scale evaluation needs, or no access to GPT-4.
>
> To summarize, the development of Auto-J extends well beyond traditional notions of novel design, such as new algorithms, metrics, or architectures. We pay special attention to a more realistic and user-centric perspective on the field of practical LLM evaluation concerning their alignment with human needs, and this is the solid contribution we want to emphasize.
>
> ---
>
> References:
>
> [1] Judging LLM-as-a-Judge with MT-Bench and Chatbot Arena, in NeurIPS 2023
>
> [2] LIMA: Less Is More for Alignment, in NeurIPS 2023
>
> [3] FLASK: Fine-grained Language Model Evaluation based on Alignment Skill Sets, Arxiv 2307.10928
>
> [4] Measuring Progress on Scalable Oversight for Large Language Models, Arxiv 2211.03540
>
> [5] PandaLM: An Automatic Evaluation Benchmark for LLM Instruction Tuning Optimization, Arxiv 2306.05087

---

> ### Author Response · Authors · 2023-11-17
> **Response to Weakness 2 and Question 1**
>
> # Response to Weakness 2
>
> ---
>
> > “The presentation of the method and contribution feels very confusing to me (maybe it's just my fault). See questions below. I do not know whether other reviewers would have similar concerns though.”
>
> We apologize for making you confused. We have made our responses to your questions below, and we have updated the new experimental results on Auto-J’s generality on the unseen scenarios in our paper (Appendix C). We hope our responses below can answer your questions / address your concerns.
>
> # Response to Question 1
>
> ---
>
> > “If I understand correctly, did you define the scenarios and categories both for fine-tune the model to perform evaluation AND test the model to perform evaluation? In that case, if we want to evaluate an LM that performs a new task that is not in the 58 categories you have defined, would Auto-J generalize to categories unseen during training?”
>
> Thanks for this valuable question on Auto-J’s generality on unseen scenarios. Yes, for both training and testing, we adopt data from all the 58 defined scenarios. Actually, in designing these scenarios, we have deliberately defined an “others” scenario for all possible unseen scenarios in testing.
>
> We also admit that the generality of Auto-J needs to be further studied, we thus conduct two extra experiments:
> We randomly select one scenario from each group as the unseen scenarios, and retrain Auto-J with the remaining scenarios.
> We take the complete “NLP tasks” group as the unseen scenarios,
> and retrain Auto-J with the remaining scenarios.
> Under both settings, we compare the new models with the complete version of Auto-J.
>
> The results are as follows, we have also updated our paper with these results in Appendix C:
>
> **Setting 1: Select one scenario from each group to form an unseen set, in total 8 unseen scenarios.**
>
> Pairwise comparison task (agreement to human preference):
>
> |Model							|seen		|unseen	|complete|
> |-|-|-|-|
> |Auto-J (paper’s version) 				|54.5		|56.8		|54.8|
> |Auto-J (remove unseen scenarios in training)	|53.5		|55.7		|53.8|
>
> Critique generation task (win rate against ChatGPT judged by GPT4):
>
> |Model							|seen		|unseen	|complete|
> |-|-|-|-|
> |Auto-J (paper’s version)				|146/200	|25/32		|171/232|
> |Auto-J (remove unseen scenarios in training) 	|143/200	|25/32		|168/232|
>
> **Setting 2: Take the complete “NLP tasks” group as the unseen scenarios, in total 11 unseen scenarios.**
>
> Pairwise comparison task (agreement to human preference):
>
> |Model							|seen		|unseen	|complete|
> |-|-|-|-|
> |Auto-J (paper’s version)				|54.2		|57.6		|54.8|
> |Auto-J (remove unseen scenarios in training)	|54.2		|54.9		|54.3|
>
> Critique generation task (win rate against ChatGPT judged by GPT4):
>
> |Model							|seen		|unseen	|complete|
> |-|-|-|-|
> |Auto-J (paper’s version) 				|136/188	|35/44		|171/232|
> |Auto-J (remove unseen scenarios in training) 	|130/188	|38/44		|168/232|
>
> Compared with the complete version of Auto-J, the two re-trained variants only show slightly degraded performance on the two evaluated tasks both on the seen and unseen scenarios, which indicates that Auto-J can generalize well to scenarios unseen during training.

---

> ### Author Response · Authors · 2023-11-17
> **Response to Question 2**
>
> # Response to Question 2
>
> ---
>
> > “Did you use GPT-4 as the label generator?”
>
> Our training data consists of two parts: pairwise comparison and single response evaluation.
>
> In collecting pairwise comparison data, we do not rely on GPT-4 solely. We utilize existing human preference annotation verdicts (indicating which response is preferred) to filter out those samples where GPT-4 disagrees with humans. So you can regard it like this: the verdict labels are from humans, and GPT-4 only adds the explanations for these labels.
>
> In collecting single response evaluation, since we do not have similar human labels as in pairwise comparison data, we generate all data from GPT-4, including the critiques and the overall ratings for responses.
>
> > “If so, if one had enough OpenAI credit, the optimal strategy of evaluation according to this paper would still be using GPT-4 or not?”
>
> The answer to this question is almost yes, but relies on certain conditions. While GPT-4 is indeed powerful, harnessing its full potential for evaluating other models requires a substantial degree of prompt engineering skill, which may be challenging for average users. In contrast, Auto-J offers a 'ready-to-use' solution that doesn't necessitate an in-depth grasp of the models to be evaluated or the specific evaluation scenario, thanks to the pre-encoded information (like evaluation criteria) during its training phase.
>
> Besides, we also want to emphasize that Auto-J is not designed to completely substitute GPT-4, but to offer an open-source and efficient lightweight alternative for scenarios with limited resources and budget or even do not have access to GPT4:
>
> - Auto-J can process 100 samples per minute on an A100 GPU, which in our practice GPT-4 can only process 4 samples per minute.
>
> - Auto-J has only 13B parameters, which means it can be deployed on 32G V100 GPUs (FP16), and after quantization it can even be used on commodity-level GPUs like Nvidia 4090 or 3090.
>
> - Finally, Auto-J has quite good performance. On pairwise comparison tasks, it only lags behind GPT-4 and surpasses other competitive baselines like ChatGPT or Claude-2 significantly (Table 1). On single response evaluation, it even beats GPT-4 (Figure 3).

---

> > ### Author Response · Authors · 2023-11-21
> >
> > Dear Reviewer PG92,
> >
> > We recognize that the timing of this discussion period may not align perfectly with your schedule, yet we would greatly value the opportunity to continue our dialogue before the deadline approaches.
> >
> > Could you let us know if your concerns have been adequately addressed? If not, please feel free to raise them, and we are more than willing to provide further clarification; if you find that your concerns have been resolved, we would appreciate if you could re-consider the review score.
> >
> > We hope that we have resolved all your questions, but please let us know if there is anything more.
> >
> > Thanks.

---

### Official Review · Reviewer_WSEj · 2023-10-31

**Soundness:** 3 good
**Presentation:** 2 fair
**Contribution:** 2 fair
**Rating:** 5
**Confidence:** 3

**Summary:**

This paper aims to present Auto-J, an LLM that may evaluate LLMs. This is in the line of LMs that are used to evaluate other LMs.

**Strengths:**

- Auto-J proposes a way to produce evaluation methods for LLMs

**Weaknesses:**

- It is strange that larger models are used to evaluate other models
- LLMs should somehow emulate human capabilities and not other LLMs' capabilities.

**Questions:**

It is clear that this paper is in a long line of other approaches. Yet, it is not clear why evaluating LLMs with an LLM is principled. Cna you comment on this?

---

> ### Author Response · Authors · 2023-11-17
> **Response to Weakness 1**
>
> # Response to Weakness 1
>
> ---
>
> > “It is strange that larger models are used to evaluate other models”
>
> Thanks for your feedback. We would like to offer additional context to the point you raised regarding the use of larger models for evaluating outputs in text generation, as it is indeed a prevalent practice within the field.
>
> **Pre-trained Model Based Evaluation**: Long before the wide recognition of Large Language Models (LLMs), we have seen many examples (including a survey[1]) like BERTScore [2], BLEURT [3], RoBERTa-Eval [4], BARTScore [5] which rely on pre-trained language models like BERT, RoBERTa or BART in evaluating text generation. These methods show better correlation with human judgments than metrics like BLEU or ROUGE on many text generation tasks like text summarization or machine translation, because the pre-trained models are able to capture more semantic information than text-matching methods.
>
> **LLM-based Evaluation**: With the development of LLMs, this trend continues because powerful LLMs have been deployed to a much wider range of real-world scenarios. The outputs are becoming much longer and more complex, and these tasks usually have no ground truth response as a reference. Therefore, using larger models to evaluate other models has become an effective and almost the only solution for automatic evaluation because of the massive knowledge stored in their parameters, the strong reasoning, and the instruction-following ability. Many works have adopted this paradigm in assessing various models and building standard benchmarks, like AlpacaEval [6] or MTBench [7].
>
> Along this research line, we develop Auto-J to ease the heavy reliance on expensive and closed-source LLMs like GPT-4 or ChatGPT in evaluation, and we have shown with many experiments that Auto-J is comparable to them on performance and with a significant advantage on cost, speed, and resource-saving.
>
> We hope our detailed explanation contributes to a deeper understanding of the context and addresses any concerns you may have. We also welcome further discussions on this topic.
>
> ---
>
> References:
>
> [1] A Survey of Evaluation Metrics Used for NLG Systems, in ACM Computing Surveys 2022
>
> [2] BERTScore: Evaluating Text Generation with BERT, in ICLR 2020
>
> [3] BLEURT: a Transfer Learning-Based Metric for Natural Language Generation, in ACL 2020
>
> [4] Designing Precise and Robust Dialogue Response Evaluators, in ACL 2020
>
> [5] BARTScore: Evaluating Generated Text as Text Generation, in NeurIPS 2021
>
> [6] AlpacaFarm: A Simulation Framework for Methods that Learn from Human Feedback, in NeurIPS 2023
>
> [7] Judging LLM-as-a-Judge with MT-Bench and Chatbot Arena, in NeurIPS 2023

---

> ### Author Response · Authors · 2023-11-17
> **Response to Weakness 2**
>
> # Response to Weakness 2
>
> ---
>
> > “LLMs should somehow emulate human capabilities and not other LLMs' capabilities.”
>
> Thanks for this valuable comment. We would like to offer additional context to the point you raised regarding training a model to mimic another LLM’s capabilities.
>
> **Previous works that learn from AI-generated data/feedback**: Actually, modern LLMs like GPT3.5 or GPT-4 have become so powerful that they can serve as an excellent crowdsourced annotator by providing them with sufficient guidance and demonstrated examples [1, 2]. Such an idea of learning from data generated from the combination of the strong text generation ability of LLMs and carefully designed human prior/supervision (like data filtering, manual seed tasks, provided demonstrations or careful prompt engineering) has been widely adopted in recent works like Alpaca [3], Vicuna [4], WizardLM [5], Zephyr [6] and Prompt2Model [7], and are able to produce models with comparable performance with proprietary models like ChatGPT.
>
> There are also works that make models to learn from AI feedback, like Constitutional AI [8] or RLAIF [9]. These works have demonstrated the effectiveness of learning from AI (e.g. a powerful LLM), and are comparable to methods that learn from human feedback with regard to helpfulness and harmlessness.
>
> **Our work is not simply emulating GPT-4**: Our work builds upon the concepts of the aforementioned studies, harnessing the powerful text generation capabilities of GPT-4 under a series of carefully designed guides (such as defined scenarios and manually crafted evaluation criteria) to obtain more comprehensive and specific generated judgments for model training. This is not merely an imitation of GPT-4’s abilities; instead, GPT-4 is utilized as an auxiliary annotation tool that can significantly reduce human labor. Furthermore, we employ various other processes, such as filtering out samples where GPT-4’s responses conflict with existing human labels and employing a "divide-and-conquer" strategy in the generation of single response evaluation data. These techniques ensure that the capabilities of the model we develop are not confined to the simple direct prompting of GPT-4’s upper limits of performance.
>
> By all these techniques, we have successfully developed Auto-J and show with extensive experiments that it surpasses all baselines except GPT-4 in pairwise response comparison (Table 1) and even outperforms GPT-4 in critique generation (Figure 3), using fewer than 5,000 training samples, which demonstrates high data efficiency of our method.
>
> We hope our elaboration can contribute to a deeper understanding of the context and address your concern. We are also welcome to any further discussions on this topic.
>
> ---
>
> References:
>
> [1] AnnoLLM: Making Large Language Models to Be Better Crowdsourced Annotators, Arxiv 2303.16854
>
> [2] ChatGPT outperforms crowd-workers for text-annotation tasks, Arxiv 2303.15056
>
> [3] Stanford Alpaca: An instruction-following llama model, https://github.com/tatsu-lab/stanford_alpaca
>
> [4] Vicuna: An open-source chatbot impressing gpt-4 with 90%* chatgpt quality,
> https://lmsys.org/blog/2023-03-30-vicuna/
>
> [5] WizardLM: Empowering large language models to follow complex instructions, Arxiv 2304.12244
>
> [6] Zephyr: Direct Distillation of LM Alignment, Arxiv 2310.16944
>
> [7] Prompt2Model: Generating Deployable Models from Natural Language Instructions, Arxiv 2308.12261
>
> [8] Constitutional AI: Harmlessness from AI Feedback, Arxiv 2212.08073
>
> [9] RLAIF: Scaling Reinforcement Learning from Human Feedback with AI Feedback, Arxiv 2309.00267

---

> ### Author Response · Authors · 2023-11-17
> **Response to Question 1**
>
> # Response to Question 1
>
> ---
>
> > “It is clear that this paper is in a long line of other approaches. Yet, it is not clear why evaluating LLMs with an LLM is principled. Can you comment on this?”
>
> As we have stated in the response to Weakness 1, the application of current LLMs is much broader than that of previous task-specific models. They may be deployed in many scenarios where there are no ground truth responses for each task, and the outputs can be challenging for average crowdsourced workers to judge. [1]
>
> The optimal and ideal way to evaluate the outputs of LLMs would be to ask in-domain human experts to judge these texts. However, that approach can be expensive and yield inconsistent results.
>
> On the other hand, many previous studies have demonstrated the effectiveness of using one LLM to judge another. These works [2,3] have provided sufficient experimental results to empirically show that the judgments of LLMs highly agree with human judgments. There is also work [4] that shows with the help of a trained critique model, humans can find more flaws in the outputs of other models.
>
> Also, using powerful LLMs (like GPT-4) to evaluate other LLMs enables the building of a benchmark to automatically show how different LLMs perform on a massive array of real-world user queries.
>
> So, in conclusion, evaluating LLMs with an LLM is principled and reasonable, both from the actual need to evaluate LLMs' diverse and unstandardized output in real-world settings and from the vast amount of empirical experimental results shown in previous studies.
>
> ---
>
> References:
>
> [1] Measuring Progress on Scalable Oversight for Large Language Models, Arxiv 2211.03540
>
> [2] AlpacaFarm: A Simulation Framework for Methods that Learn from Human Feedback, in NeurIPS 2023
>
> [3] Judging LLM-as-a-Judge with MT-Bench and Chatbot Arena, in NeurIPS 2023
>
> [4] Self-critiquing models for assisting human evaluators, Arxiv 2206.05802

---

> > ### Author Response · Authors · 2023-11-21
> >
> > Dear Reviewer WSEj,
> >
> > We recognize that the timing of this discussion period may not align perfectly with your schedule, yet we would greatly value the opportunity to continue our dialogue before the deadline approaches.
> >
> > Could you let us know if your concerns have been adequately addressed? If not, please feel free to raise them, and we are more than willing to provide further clarification; if you find that your concerns have been resolved, we would appreciate if you could re-consider the review score.
> >
> > We hope that we have resolved all your questions, but please let us know if there is anything more.
> >
> > Thanks.

---

### Official Review · Reviewer_jzn2 · 2023-11-01

**Soundness:** 3 good
**Presentation:** 3 good
**Contribution:** 2 fair
**Rating:** 5
**Confidence:** 4

**Summary:**

This paper proposes a generative judge with 13 billion parameters to evaluate the generations of large language models from real-world scenarios. Specifically, the authors created a large collection of data among 58 different scenarios and guided GPT4 to collect evaluation judgements as supervised training signals. Extensive evaluations demonstrate Auto-J outperforms many strong baselines and more analysis shows advantages of the proposed method, like reducing positional bias and generating more specific critiques.

**Strengths:**

1). The design of scenario-specific criteria is strongly motivated, which will enable LLM-based judges to produce high-quality evaluations and critiques. Curated criterias can be model-agnostic and adopted to multiple models.

2). Comprehensive evaluation and analysis of Auto-J demonstrate that its evaluations are consistent and can align well with human judgements.

**Weaknesses:**

The technical contribution is a bit limited as it is still within the scope of training one more LLM as judges to evaluate other LLMs’ generation. Since the training data is obtained from GPT4’s output, it is unsure whether it can replace GPT4 as judges or has strong generalizations as GPT4.

**Questions:**

1). Is it necessary to have a 13b parameter model to train the scenario classifier? Have you tried other simple BERT-like models?

2).Is there any bad case that Auto-J fails or Auto-J generates wrong critiques?

**Details Of Ethics Concerns:**

n.a.

---

> ### Author Response · Authors · 2023-11-17
> **Response to Weakness 1**
>
> # Response to Weakness 1
>
> ---
>
> Thank you for your valuable feedback. We would like to use this opportunity to clarify the scope of the technical contributions in our work and further detail the efforts and design principles behind our data collection process.
>
> > Concern on the limited technical contribution: “The technical contribution is a bit limited as it is still within the scope of training one more LLM as judges to evaluate other LLMs’ generation.”
>
> **The focus shift in LLM generation evaluation**: In this paper, we focus on the field of evaluating Large Language Models (LLMs) on how they align with human needs in massive real-world scenarios. This field has garnered increasing concern due to the growing capabilities of LLMs and their wider deployment across various scenarios.
>
> As our introduction notes, the field has been pivoting away from a narrow definition of technical contribution—such as designing specific algorithms, metrics, or model architectures for evaluators—toward a more user-centric evaluation approach. This shift reflects a collective recognition that the true measure of an LLM's utility is not solely captured by its performance under standard academic benchmarks but rather by its effectiveness in diverse, real-world contexts [1, 2, 3].
>
> **Our contribution under the practical necessity**: In response to the actual needs in this research field, we propose Auto-J, which demonstrates better evaluation capabilities than many competitive baselines with significantly more parameters (such as ChatGPT, Claude2, LLaMA-2-70B-Chat). Since Auto-J achieves these results with only 13B parameters, it also lays a solid foundation for the research line of scalable oversight [4]: by combining human efforts with model capabilities, a small model can serve as a good overseer for much stronger models, ensuring effective oversight even if a model's abilities surpass human capabilities.
>
> > Concern on using data generated by GPT4 for training: “Since the training data is obtained from GPT4’s output, it is unsure whether it can replace GPT4 as judges or has strong generalizations as GPT4.”
>
> **Carefully designed human-GPT4 collaboration in data collection**: We want to emphasize that we do not rely solely on GPT-4 for constructing the training data. Instead, we employ a synergistic blend of GPT-4's advanced capabilities with meticulously crafted human inputs (such as scenario definitions and criteria design). The data compilation process is heavily reliant on human labor and ingenuity, encompassing finding the sources for user inquiries and potential model responses.
>
> We have also handcrafted a large collection of evaluation criteria, which effectively guide GPT-4 to generate more comprehensive and specific judgments than direct prompting without any guidance. We employ data filtering (using existing human labels) in pairwise comparison data collection and execute a “divide-and-conquer” strategy (combining judgments with and without reference to the criteria) in single response evaluation data collection.
>
> **Good performance with data efficiency**: With all the above-mentioned techniques, we have successfully developed Auto-J in a data-efficient manner with less than 5,000 training samples. We show with extensive experiments that Auto-J surpasses all baselines except GPT-4 in pairwise response comparison (Table 1) and even outperforms GPT-4 in critique generation (Figure 3).
>
> **Evaluator under low-resource settings**: Furthermore, although GPT-4 may perform better when resources (like time and budget) are ample, Auto-J is open-sourced, smaller, and faster.
> Auto-J, by design, is not expected to be viewed as a “neither here nor there” substitute for GPT-4 but rather as a valuable complement for those with limited resources, large-scale evaluation needs, or no access to GPT-4.
>
> To summarize, the development of Auto-J extends well beyond traditional notions of technical contributions, such as new algorithms, metrics, or architectures. It represents a more realistic and user-centric perspective on the field of practical LLM evaluation concerning their alignment with human needs, and this is the solid contribution we want to emphasize.
>
> ---
>
> References:
>
> [1] Judging LLM-as-a-Judge with MT-Bench and Chatbot Arena, in NeurIPS 2023
>
> [2] LIMA: Less Is More for Alignment, in NeurIPS 2023
>
> [3] FLASK: Fine-grained Language Model Evaluation based on Alignment Skill Sets, Arxiv 2307.10928
>
> [4] Measuring Progress on Scalable Oversight for Large Language Models, Arxiv 2211.03540

---

> ### Author Response · Authors · 2023-11-17
> **Response to Question 1**
>
> # Response to Question 1
>
> ---
>
> > “Is it necessary to have a 13b parameter model to train the scenario classifier? Have you tried other simple BERT-like models?”
>
> Thank you for this insightful question. Although pursuing a perfect scenario classifier is not the central aim of this paper (we only want to train a robust tool with strong generalization capabilities for scenario classification within our training framework), we still have included a rationale for this selection and carried out additional experiments in response. We hope that the explanations and results presented hereafter can answer your question/address your concern.
>
> **Some users’ queries are quite lengthy**: The motivation for training a 13B scenario classifier is that we observe that some users’ queries are quite long and complex (with more than 1k words), while traditional BERT-like models only have a context length of 512 tokens. Training scenario classifiers based on these small models means that the input queries need to be truncated and some information may be dropped.
>
> **A larger model yields greater accuracy in classification**: We have trained two new scenario classifiers based on BERT-base and BERT-large. The results of our held-out test set are as follows:
>
> | Base Model 							| Acc 	| F1 |
> |-|-|-|
> | BERT-base 							| 39.97 | 27.02 |
> | BERT-large 							| 59.07 | 52.72 |
> | LLaMA-2-13b (scenario classifier mentioned in the paper)	| 72.55 | 74.12 |
>
> The results indicate that employing smaller models results in a marked decrease in scenario identification accuracy. Given that our scenario criteria are meticulously defined, we posit that a more sophisticated and powerful scenario classifier is essential to accurately assign the appropriate criteria to each user query. This, in turn, would enhance the quality of judgments generated by GPT-4, thereby improving the training data for our model.
>
> **Potential concern on inference efficiency**: It is also worth noting that with the help of vllm (an LLM inference acceleration framework) [1], even a 13B scenario classifier can infer very fast (more than 3k samples per minute). Since it is only used in the data collection stage, but not in the judgment generation process of Auto-J, it will not bring inference delay in actual use.
>
> ---
>
> References:
>
> [1] Efficient Memory Management for Large Language Model Serving with PagedAttention, in SOSP 2023

---

> ### Author Response · Authors · 2023-11-17
> **Response to Question 2**
>
> # Response to Question 2
>
> ---
>
> > “Is there any bad case that Auto-J fails or Auto-J generates wrong critiques?”
>
> Since the design and development of Auto-J are geared towards real-world scenarios, we pay close attention to failure cases identified by users in actual use. From the external user feedback we have received, we noticed that Auto-J falls short in criticizing the responses to complex mathematical problems (which may stem from the inherent limitations of LLaMA-2, its base model), and we provide an example below.
>
> Owing to the data efficiency, the lightweight nature of Auto-J, and the proposed scalable data synthesis framework (see Section 3), we are able to rapidly iterate new versions of Auto-J. After observing its deficiency in math, we have collected more data on mathematics, which we have integrated with the previous training data to train a new version of the model.
>
>
> **A failure case of Auto-J**:
>
> User’s query:
> ```
>     Humans continuously think, they may say a few words, but in their thoughts in a couple of sentences. Do the same. Put your thoughts in parentheses like ("your thoughts"). You do this as frequently as possible.
>
> 	Now solve this: x^2 - 8 *x + 12 = 0, using thoughts.
> ```
>
> Response:
> ```
>     Sure, I'll do my best to solve the quadratic equation using thoughts. Here's my thought process:
> 	1. First, I'll substitute x with the expression inside the parentheses: x = 6.
> 	2. Now I'll calculate the discriminant using the formula: (b^2) - 4ac. In this case, a = 1, b = -8, and c = 12.
> 	   discriminant = (-8)^2 - 4 * 1 * 12
> 	   discriminant = 64 - 64
> 	   discriminant = 0
> 	3. Now I'll plug the discriminant back into the quadratic formula: x = (-8 ± 0) / 2, or x = -4 or x = 4.
> 	4. Since we're looking for two solutions, we'll choose the positive value: x = 4.
> 	5. So the solution to the quadratic equation x^2 - 8x + 12 = 0 using thoughts is x = 4.
> ```
>
> Auto-J’s judgment (truncated):
> ```
>     The response provided to the user's query has several areas that need improvement. Firstly, the user's request for the responder to "put their thoughts in parentheses like ('your thoughts')" was not adhered to. The responder's thoughts were not clearly distinguished from the mathematical operations, making it difficult for the user to understand the responder's thought process.
>     Secondly, the responder made an error in the calculation of the discriminant. The correct calculation should be (-8)^2 - 4 * 1 * 12 = 64 - 48 - 12 = -10, not 64 - 64 = 0. This error, if uncorrected, would lead to incorrect solutions for the quadratic equation.
>     …
> ```
>
> We can see that although Auto-J correctly identifies the mistakes in calculating the discriminant, it does not give a correct result either.
>
> **Our effort in fixing this deficiency**:
>
> In response to Auto-J’s inability in math reasoning. We further collect 1000 pairwise response comparison samples and 1000 single response evaluation samples from the GSM8K dataset. We combine these data with the already collected data to train a new version of Auto-J.
>
> We have also built a new test set with 500 samples for the pairwise response comparison task on math reasoning questions (requires the models to decide which response is better for a math question, like providing a correct answer or a more detailed solution process). We report the performance on both the general purpose test and the math problem test for both versions of Auto-J here:
>
> |Model 				| General Testset 	| Math Testset |
> |-|-|-|
> |Auto-J (paper’s version) 	| 54.8	 		| 36.0 |
> |Auto-J (math enhanced) 	| 53.6 			| 88.0 |
>
> We see that the new version of Auto-J significantly improves its ability to criticize math reasoning problems, and still keeps a similar performance on the general purpose pairwise test set in paper.
>
> From this use case, we can see that the rapid iteration and enhancement of our model Auto-J (within a month of our submission) can be attributed to several factors:
>
>   - (1) real-world application-oriented and open-sourced, allowing for active feedback from real-world users.
>
>   - (2) a flexible data synthesis framework.
>
>   - (3) effective training methodologies.
>
> As also mentioned previously, these elements set up the major contributions of this paper. The focus is not on magic modification of the network architecture or loss function, but rather on exploring how to achieve scalable "supervision signals" in the era of generative artificial intelligence, through collaboration between human oversights and GPT models, which (i.e., scalability) enables the fulfillment of the increasingly growing demands for the model, and through open sourcing, it achieves transparency and reproducibility.

---

> > ### Author Response · Authors · 2023-11-21
> >
> > Dear Reviewer jzn2,
> >
> > We recognize that the timing of this discussion period may not align perfectly with your schedule, yet we would greatly value the opportunity to continue our dialogue before the deadline approaches.
> >
> > Could you let us know if your concerns have been adequately addressed? If not, please feel free to raise them, and we are more than willing to provide further clarification; if you find that your concerns have been resolved, we would appreciate if you could re-consider the review score.
> >
> > We hope that we have resolved all your questions, but please let us know if there is anything more.
> >
> > Thanks.

---

### Comment · Reviewer_WSEj · 2023-11-17
**Please stop putting comments separately.**

Dear Author, you are generating tons of emails in my inbox.
Can you put your comment in a single shot?

---

### Meta-Review · Area_Chair_QLjC · 2023-12-11

**Metareview:**

This paper presents an evaluator called "Auto-J", which is built by a series of steps that distills the evaluation decisions made by GPT-4 on a diverse set of input prompts (all open source) into a 13B model (base model is LLAMA-2).  Overall, the authors show that Auto-J is a very competitive evaluator of diverse phenomena that users can prompt a model with.

Strength:  I think that this paper produces a strong artifact in the form of open source data and a methodology that creates resources that can be reused by the community.  The Auto-J evaluator could be used by the research community who focus on open source LLMs.  This is a significant strength of the paper.

Weaknesses:  The reviewers point this out somewhat--but in brief, the paper reads like a technical report where the authors describe a clean pipeline to produce the Auto-J evaluator.  The reviewers say that they don't really find this to be very novel since no significantly novel idea is proposed as auto-evaluators of this kind are not uncommon.  While I agree with this to some extent, I think the merit of the paper lies in releasing artifacts that could be quite useful to the research community that is focused on developing open source LLMs.  Hence, I don't find these comments that criticize the paper to be significant.

**Justification For Why Not Higher Score:**

I do think that the paper does not present anything massively significant in terms of new science, but the overall flow of the methods in the paper present a novel artifact in terms of the Auto-J evaluator that could be useful to the community.  Given the lack of significant novelty, I am suggesting that we accept the paper as a poster.

**Justification For Why Not Lower Score:**

Please see above.  I think the paper presents some good ideas and a new set of open source artifacts that deserve acceptance.

---

### Decision · Program_Chairs · 2024-01-16

Accept (poster)